# Accelerating Diffusion Models for Inverse Problems through Shortcut Sampling

## Abstract

Recently, diffusion models have demonstrated a remarkable ability to solve inverse problems in an unsupervised manner. Existing methods mainly focus on modifying the posterior sampling process while neglecting the potential of the forward process. In this work, we propose **S**hortcut **S**ampling for **D**iffusion (**SSD**), a novel pipeline for solving inverse problems. Instead of initiating from random noise, the key concept of SSD is to find the "Embryo", a transitional state that bridges the measurement image $y$ and the restored image $x$. By utilizing the "shortcut" path of "input-Embryo-output", SSD can achieve precise restoration with reduced steps. To obtain the Embryo in the forward process, we propose Distortion Adaptive Inversion (DA Inversion). Moreover, we apply back projection as additional consistency constraints during the generation process. Experimentally, we demonstrate the effectiveness of SSD on several representative IR tasks. Compared to state-of-the-art zero-shot methods, our method achieves competitive results with only 30 NFEs. Moreover, SSD with 100 NFEs can outperform state-of-the-art zero-shot methods in certain tasks.

## 1 Introduction

Inverse problem is a classic problem in the field of machine learning. Given a low-quality (LQ) input measurement image $y$ and a forward degradation operator $H$, inverse problems aim to restore the original high-quality (HQ) image $x$ from $y = Hx + n$. Many image restoration tasks, including super-resolution (Haris et al., 2018; Wang et al., 2018), colorization (Larsson et al., 2016), inpainting (Yeh et al., 2017), deblurring (Guo et al., 2019; Zhang et al., 2018a) and denoising (Wang et al., 2022), can be considered as applications of solving inverse problems. In general, the restored image should exhibit two critical attributes: *Realism* and *Faithfulness*. The former indicates that the restored image should be of high-quality and photo-realistic, while the latter denotes that the restored image should be consistent with the input image in the degenerate subspace.

Recently, diffusion models (Sohl-Dickstein et al., 2015; Ho et al., 2020; Song et al., 2021c) have demonstrated phenomenal performance in generation tasks (Rombach et al., 2022; Dhariwal & Nichol, 2021; Xiao et al., 2022). Due to their powerful capability in modeling complex distributions, recent methods (Kawar et al., 2021; 2022; Chung et al., 2022a; 2023; Wang et al., 2023; Lugmayr et al., 2022; Song et al., 2021b) have sought to utilize pre-trained diffusion models for solving inverse problems in an unsupervised manner. These methods leverage the generative priors of pre-trained models to enhance *realism*, and enforce additional consistency constraints to ensure *faithfulness*.

Despite the successful application of existing methods in solving various inverse problems, their relatively slow sampling speed is a major drawback. Diffusion models have predefined a forward process and a generation process. Existing methods primarily concentrate on the posterior sampling $p(x|z, y)$ during the generation process while largely ignoring the potential sampling $p(z|y)$ during the forward process. Instead, these methods typically sample $z$ directly from the Gaussian prior $p(z)$. However, since the initial state of pure noise ranges far away from the target HQ images, previous methods have to travel through a long journey of sampling, typically requiring at least 100-250 neural function evaluations (NFEs), to generate the overall layout, structure, appearance and detailed texture of the restored image, and finally achieve a satisfactory result.

In this work, we propose **Shortcut Sampling for Diffusion (SSD)**, a novel pipeline for solving inverse problems in a zero-shot manner. The primary concept behind SSD is to find an appropriate

Figure 1: **Visual schematic of different framework.** (a) *Generation* starts from random noise and generates realistic but unfaithful results; (b) *DDIM Inversion* employs a deterministic inversion process and generation process, achieving faithful but unrealistic reconstruction. (c) *Previous IR methods* generate realistic and faithful results through *"Noise - Target"*, which take unnecessary steps to generate the layout and structure; (d) *SSD(ours)* adopts a shortcut-sampling path of *"Input-Embryo-Target"*, which generates realistic and faithful results with fewer steps.

transitional state, termed the *"Embryo"*, that bridges the gap between the input image $y$ and the target restored image $x$. By employing a shortcut path of *"Input - Embryo - Target"* instead of previous *"Noise-Target"*, SSD enables precise and fast restoration. **For convenience, we use the symbol $\mathscr{E}$ to refer to *"the Embryo"*.**

For the inversion process(*Input image - Embryo*), we found that simply adding random noise damages information from the input image and causes realistic yet unfaithful results. While a deterministic process such as DDIM Inversion(Song et al., 2021a), tends to generate unrealistic outcomes as illustrated in Fig. 1 (b). To address this dilemma, we introduce Distortion Adaptive Inversion (DA Inversion). By adding a controllable random disturbance at each inversion step, DA Inversion is capable of obtaining $\mathscr{E}$ that adheres to the predetermined noise distribution while preserving the majority of the input image.

For the generation process(*Embryo - Target*), we utilize the diffusion priors to generate additional details and texture, and introduce back projection technique(Tirer & Giryes, 2018; Wang et al., 2023) as additional consistency constraints. More specifically, we add a projection step after each denoising step to project the rough restored image onto the degenerate subspace, and obtain a revised version of the restored image which is consistent with the input image in the degenerate subspace. We further propose SSD$^+$ to extend the applicability to scenarios with unknown noise and more intricate degradation conditions.

To verify the effectiveness of SSD, we conduct experiments on various inverse problems, including super-resolution, colorization, inpainting and deblurring on CelebA (Karras et al., 2017) and ImageNet (Deng et al., 2009). Experiments demonstrate that SSD achieves competitive results when compared to state-of-the-art zero-shot methods (with 100 NFEs), despite utilizing only 30 NFEs. Moreover, we find that SSD with 100 NFEs can surpass state-of-the-art methods in certain IR tasks.

## 2 RELATED WORKS

### 2.1 DIFFUSION MODELS

**Denoising Diffusion Probabilistic Models** Diffusion models (Sohl-Dickstein et al., 2015; Ho et al., 2020; Song et al., 2021c) are a family of generative models that are used to model complex probability distributions of high-dimensional data. Denoising Diffusion Probabilistic Models(DDPM) (Ho et al., 2020) comprise both a forward process and a generation process. In the forward process, an image $x_0$ is transformed into Gaussian noise $x_T \sim \mathcal{N}(0, 1)$ by gradually adding random noise over T steps. We can describe each step in the forward process as:

$$x_t = \sqrt{1 - \beta_t} x_{t-1} + \sqrt{\beta_t} \epsilon, \epsilon \sim \mathcal{N}(0, 1) \tag{1}$$

where $x_t{}_{t=0}^{T}$ is the noisy image at time-step t, $\beta_t{}_{t=0}^{T}$ is the predefined variance schedule. Using reparameterization tricks (Kingma & Welling, 2013), The resulting noisy image $x_t$ can be expressed as:

$$x_t = \sqrt{\alpha_t} x_0 + \sqrt{1 - \alpha_t} \bar{\epsilon}_t, \bar{\epsilon}_t \sim \mathcal{N}(0, 1) \tag{2}$$

where $\alpha_t = \prod_{i=1}^{t}(1 - \beta_i)$. The generation process transforms gaussian noise $x_T$ to image $x_0$, the transition from $x_t$ to $x_{t-1}$ can be expressed as:

$$x_{t-1} = \frac{1}{\sqrt{1 - \beta_t}}(x_t - \frac{\beta_t}{\sqrt{1 - \alpha_t}}\epsilon_\theta(x_t, t)) + \frac{1 - \alpha_{t-1}}{1 - \alpha_t}\beta_t \tag{3}$$

and $\epsilon_\theta(x_t, t)$ is a neural network trained to predict the noise $\epsilon$ from noisy image $x_t$ at time-step t. The noise approximation model $\epsilon_\theta(x_t, t)$ can be trained by minimize the following objective:

$$\min_\theta \mathbb{E}_{x_0 \sim q(x_0), \epsilon \sim N(0, I)} \|\epsilon - \epsilon_\theta(x_t, t)\|_2^2 \tag{4}$$

**Denoising Diffusion Implicit Models**   Meanwhile, Song et al. (2021a) generalize DDPM via a non-Markov diffusion process that shares the same training objective, whose generation process is outlined as follows:

$$x_{t-1} = \sqrt{\alpha_{t-1}}f_\theta(x_t, t) + \sqrt{1 - \alpha_{t-1} - \sigma_t^2}\epsilon_\theta(x_t, t) + \sigma_t\epsilon_t \tag{5}$$

where $f_\theta(x_t, t)$ is the prediction of clean image $x_0$ at time-step $t$:

$$f_\theta(x_t, t) = \frac{x_t - \sqrt{1 - \alpha_t}\epsilon_\theta(x_t, t)}{\sqrt{\alpha_t}} \tag{6}$$

When $\sigma_t = 0$, DDIM samples images through a deterministic generation process, which allows for high-quality sampling with fewer steps:

$$x_{t-1} = \sqrt{\alpha_{t-1}}f_\theta(x_t, t) + \sqrt{1 - \alpha_{t-1}}\epsilon_\theta(x_t, t) \tag{7}$$

## 2.2 SOLVING INVERSE PROBLEMS IN A ZERO-SHOT WAY

A general inverse problem aims to restore a high-quality image $x$ from a known degradation operator $H$ and the degraded measurement $y$ with random additional noise $n$:

$$y = Hx + n \tag{8}$$

Traditional IR methods typically train an end-to-end model to learn the posterior $p(x|y)$ for specific tasks. While end-to-end methods can effectively restore images with in-domain degradation, they usually yield poor performance in the face of out-of-domain degradation. Meanwhile, some methods investigate leveraging the generative priors of pre-trained generative models to restore degraded images in a zero-shot way. GAN Inversion aims to find the closest latent vector in the GAN space for an input image(Xia et al., 2022; Ulyanov et al., 2018; Pan et al., 2021; Menon et al., 2020). Using the GAN Inversion technique, PULSE(Menon et al., 2020) iteratively optimizes the latent code of pre-trained StyleGAN(Karras et al., 2019) until results are consistent with the input image.

Compared with GAN, diffusion models offer a forward process that enables the direct acquisition of latent vectors within the Gaussian noise space. By performing generation processes and using consistency constraints at each step, diffusion models can be applied to various IR problems. DDRM(Kawar et al., 2022) applies SVD to decompose the degradation operators and perform diffusion in its spectral space for various IR tasks. Repaint(Lugmayr et al., 2022) proposes to solve inpainting task by retaining the unmasked area during the generation process. MCG(Chung et al., 2022a) applies a projection-based measurement consistency step at each denoising step to achieve image restoration. DPS(Chung et al., 2023) proposes an approximation of the posterior sampling to solve nonlinear inverse problems. DDNM(Wang et al., 2023) uses range-null space decomposition to decompose the restored image as a null-space part and a range-space part, they keep the range-space part unchanged to force consistency, and obtain the null-space part through iterative refinement.

## 3 METHOD

### 3.1 SHORTCUT SAMPLING

As discussed above, diffusion models comprise two processes: a forward process which progressively adds noise to the image until complete Gaussian noise, and a generation process which generates

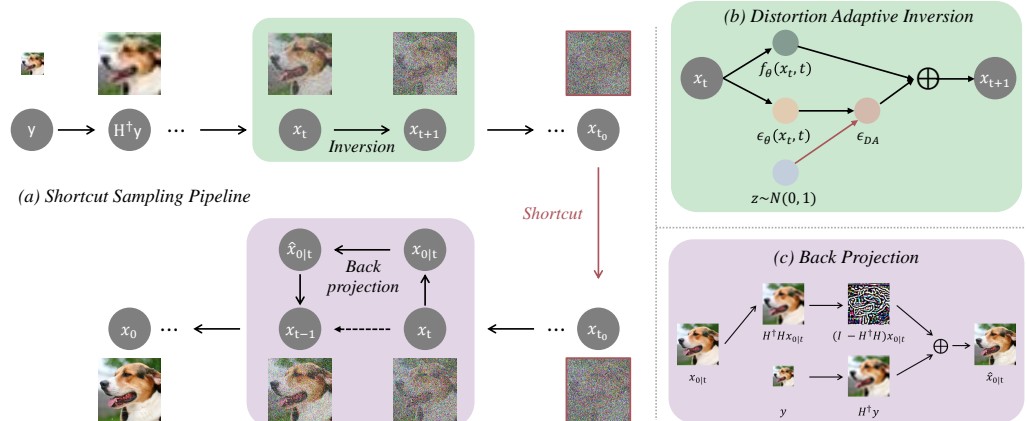

Figure 2: **Overview of the proposed SSD.** We propose a shortcut sampling pipeline, instead of starting from random noise and spending lots of steps to generate the overall layout and structure, we use Distortion Adaptive Inversion to obtain *the Embryo*, a noisy image that contains most structure information of the input images. Then during the generation process, we iteratively perform the denoising step and the back projection step to generate images with detailed texture while keeping the restored images consistent with the input images.

realistic images through iteratively denoising. Previous methods mainly focus on modifying the posterior sampling process $p(x|z, y)$ during the generation process, while ignoring the utilization of the forward process. Instead, these methods typically sample z directly from the Gaussian prior $p(z)$.

In this work, we propose Shortcut Sampling for Diffusion (SSD), a novel pipeline for solving inverse problems in a zero-shot manner. Different from previous methods that initiate from pure noise, SSD enhances the forward process to obtain an intermediate state called *the Embryo*($\mathscr{E}$), which serves as a bridge between the measurement image $y$ and the restored image $x$. Throughout the shortcut sample path of "input-$\mathscr{E}$-output", SSD can achieve efficient and satisfactory restoration results.

For convenience, we denote the transition from the measurement image $y$ to $\mathscr{E}$ as the "inversion process"; and the transition from $\mathscr{E}$ to the restored image $x$ as the "generation process". Given a LQ image $y$ and corresponding degraded operator $H$, we start from $H^\dagger y$ and apply Distortion Adaptive Inversion (DA Inversion) to derive $\mathscr{E}$ in the inversion process(Sec. 3.2). Subsequently, during the generation process, we iteratively perform the denoising step and the back projection step to generate both faithful and realistic results(Sec. 3.3). Further, due to SSD relies on an accurate estimation of degraded operators to exhibit high performance, we proposed an enhanced version called SSD$^+$ that makes SSD suitable for noisy situations or inaccurate estimation of $H$.(Sec. 3.4). The overall pipeline of SSD is illustrated in Fig. 2.

## 3.2 DISTORTION ADAPTIVE INVERSION

We expect to obtain *the Embryo* by enhancing the forward process. As previously discussed, *the Embryo* should satisfy the following criterias:

**Criteria (i)**: *The Embryo* should contain information from the input image;

**Criteria (ii)**: *The Embryo* should retain the capacity for generating a high-quality image.

**Why DDIM Inversion and DDPM Inversion Cannot Work Well** To satisfy **Criteria (i)**, a naive approach is to build a deterministic mapping from the input image $y$ to $\mathscr{E}$. Given the deterministic nature of the DDIM generation process, we can establish the DDIM Inversion process by reversing Eq. (7) in the following manner:

$$x_{t+1} = \sqrt{\alpha_{t+1}} f_\theta(x_t, t) + \sqrt{1 - \alpha_{t+1}} \epsilon_\theta(x_t, t) \qquad (9)$$

*The Embryo* obtained through DDIM Inversion preserves most information of the input images since we can reconstruct it by iteratively executing Eq. (7). However, as depicted in Fig. 3 (a), the

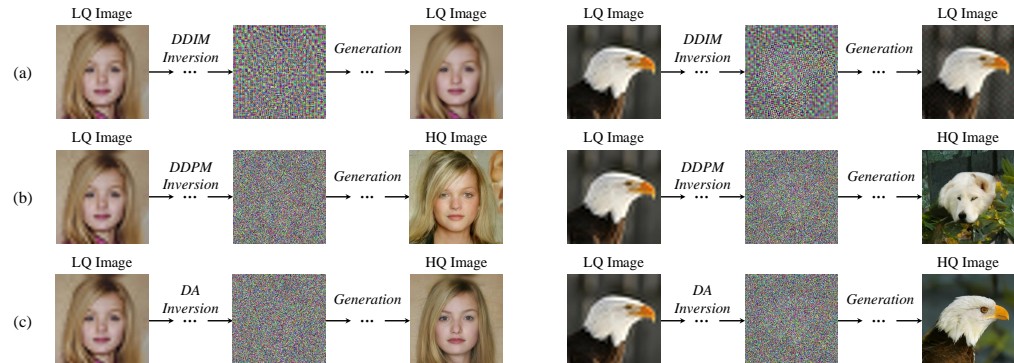

Figure 3: **Comparison of reconstruction results between different Inversion Methods. (a)** *DDIM Inversion* produces faithful but unrealistic results. **(b)** *DDPM Inversion* produces realistic but unfaithful results **(c)** *Distortion Adaptive Inversion(ours)* produces both realistic and faithful results

application of this Embryo in the generation process produces faithful but unrealistic results, thus violating **Criteria (ii)**.

We attribute the failure to the observation that, the obtained Embryo deviates from the predefined noise distribution. More specifically, given a low-quality input image $y$, the predicted noise $\{\epsilon_\theta(x_t, t)\}$ during DDIM Inversion process deviates from the standard normal distribution, resulting in the deviation of $\mathscr{E}$ from the predefined noise distribution. During the generation process, the pre-trained model receives out-of-domain input distributions, thereby generating unrealistic results. We summarize this observation as follows, more details is available in Appendix B:

**Assumption 1.** *Diffusion Models rely on in-domain noise distribution to generate high-quality images. When facing low-quality input images, the distribution of predicted noise $\epsilon_\theta(x_t, t)$ during the DDIM Inversion process exhibits a greater deviation from the standard normal distribution.*

Another extreme scenario is the random forward process, which can be regarded as a special inversion technique termed **DDPM Inversion**. In DDPM Inversion, the predicted noise is replaced with randomly sampled noise from Gaussian distribution. We can define it in a similar form of DDIM Inversion in Eq. 9:

$$x_{t+1} = \sqrt{\alpha_{t+1}} f_\theta(x_t, t) + \sqrt{1 - \alpha_{t+1} - \beta_{t+1}} \epsilon_\theta(x_t, t) + \sqrt{\beta_{t+1}} z, z \sim \mathcal{N}(0, 1) \quad (10)$$

As shown in Fig. 3(b), DDPM Inversion converts the input image $y$ into pure noise, thereby violating **Criterion (i)** and producing results that are realistic yet lack faithfulness.

**Distortion Adaptive Inversion** Since a deterministic inversion process like DDIM Inversion produces unrealistic results, while a stochastic inversion process like DDPM Inversion yields unfaithful results. To resolve this dilemma, we propose a novel inversion approach called Distortion Adaptive Inversion(DA Inversion). The definition of DA Inversion is stated as follows:

**Definition 1.** *We define the iterative process of Distortion Adaptive Inversion as:*

$$x_{t+1} = \sqrt{\alpha_{t+1}} f_\theta(x_t, t) + \sqrt{1 - \alpha_{t+1} - \eta\beta_{t+1}} \epsilon_\theta(x_t, t) + \sqrt{\eta\beta_{t+1}} z \quad (11)$$

*where $\eta$ control the proportion of random disturbances and $0 < \eta < 1$.*

*For ease of exposition, the predicted noise in DA Inversion at each time-step can be rephrased as:*

$$\epsilon_{DA} = \frac{1}{\sqrt{1 - \alpha_{t+1}}} (\sqrt{1 - \alpha_{t+1} - \eta\beta_{t+1}} \epsilon_\theta(x_t, t) + \sqrt{\eta\beta_{t+1}} z), z \sim \mathcal{N}(\mu, \sigma) \quad (12)$$

By adding controllable random perturbations in each inverse step, DA Inversion has the capability to generate high-quality images while preserving the essential information including layout and structure, which is shown in Fig. 3 (c).

For **Criterion (i)**, since the random perturbation only replaces a portion of the predicted noise, the Embryo obtained through DA Inversion actually preserves a substantial amount of information from

the input image. For **Criterion (ii)**, we have verified that incorporating random disturbances can bring the predicted noise closer to $\mathcal{N}(0,1)$. Proofs are available in Appendix A

**Proposition 1.** *Assuming $\epsilon_\theta(x_t, t) \sim \mathcal{N}(\mu, \sigma^2)$, We have:*

$$\epsilon_{DA} \sim \mathcal{N}(\frac{\sqrt{1-\alpha_{t+1}-\eta\beta_{t+1}}}{\sqrt{1-\alpha_{t+1}}}\mu, 1 + \frac{1-\alpha_{t+1}-\eta\beta_{t+1}}{1-\alpha_{t+1}}(\sigma^2-1)) \tag{13}$$

*thus:*

$$\begin{aligned} \|\mu_{\epsilon_{DA}}\| &< \|\mu\| \\ \|\sigma^2_{\epsilon_{DA}} - 1\| &< \|\sigma^2 - 1\| \end{aligned} \tag{14}$$

*which indicates that after adding random disturbance, $\epsilon_{DA}$ becomes closer to $\mathcal{N}(0,1)$.*

In practice, rather than performing the inversion process until the last time-step $T$, we find that we can achieve acceleration by performing until time-step $t_0 < T$, which is inspired by (Meng et al., 2021; Kim et al., 2022; Chung et al., 2022b).

### 3.3 BACK PROJECTION

Although $\mathcal{E}$ obtained from DA Inversion carries information about the input image, and the generation process started from which can produce images with high quality, the result may not entirely align with the input LQ image in the degenerate subspace. To address this problem, we introduce the back projection technique as consistency constraints during the generation process.

Back projection was originally introduced by Tirer & Giryes (2018) to solve inverse problems. Previous works(Kawar et al., 2022; Wang et al., 2023) have also utilized back projection in pretrained diffusion models as additional consistency constraints. For details, given a measurement image y and corresponding degraded operator $H$, we can project the roughly restored image $x$ onto the affine subspace $\{H\mathbb{R}^n = y\}$ by the following operation to force consistency:

$$x' = (I - H^\dagger H)x + H^\dagger y \tag{15}$$

The refined restored image $x'$ consists of two parts: the former part, denoted as $(I - H^\dagger H)x$, represents the residual between $x$ and the image obtained after projection-in and projection-back, which can be interpreted as the enhancement details of $x$. The latter part, denoted as $H^\dagger y$, can be regarded as the preservation of input image $y$. Following the back-projection step, we have:

$$Hx' \equiv H[(I - H^\dagger H)x + H^\dagger y] \equiv y \tag{16}$$

which indicates that $x'$ entirely aligns with the input measurement image $y$ in the degenerate subspace.

Inspired by prior research, we incorporate back projection to enhance consistency. At each time-step, we initiate the process by performing a denoising step to obtain the predicted $x_0$ in Eq. 6. Subsequently, we process with a back projection step to refine the results of $x_0$ and derive $x_{t-1}$ through Eq. 5. The complete transformations from $x_t$ to $x_{t-1}$ can be expressed as follows:

$$\begin{aligned} x_{0|t} &= \frac{x_t - \sqrt{1-\alpha_t}\epsilon_\theta(x_t, t)}{\sqrt{\alpha_t}} \\ \hat{x}_{0|t} &= (I - H^\dagger H)x_{0|t} + H^\dagger y \\ x_{t-1} = \sqrt{\alpha_{t-1}}\hat{x}_{0|t} &+ \sqrt{1-\alpha_{t-1}-\sigma_t^2}\epsilon_\theta(x_t, t) + \sigma_t\epsilon_t \end{aligned} \tag{17}$$

### 3.4 EXPAND SSD TO NOISY IR TASKS

Although SSD is effective in addressing various noiseless inverse problems, it tends to exhibit poor performance when faced with noisy tasks. This limitation can be primarily attributed to the utilization of back projection. The success of back projection hinges on a precise estimation of the degraded operator $H$. When applied to blind image restoration or noisy IR tasks, back-projection tends to result in disappointing restorations because of the inability to satisfy Eq. 16.

To solve this problem, we proposed SSD$^+$, an enhanced version that makes SSD suitable for noisy situations or inaccurate estimation of $H$. Earlier studies(Meng et al., 2021; Hertz et al., 2022;

| CelebA | SR × 4 | SR × 8 | Colorization | Deblur (gauss) | NFEs↓ |
|---|---|---|---|---|---|
| Method | PSNR↑ / FID↓ / LPIPS↓ | PSNR↑ / FID↓ / LPIPS↓ | FID↓ / LPIPS↓ | PSNR↑ / FID↓ / LPIPS↓ | |
| $H^†y$ | 28.02 / 128.22 / 0.301 | 24.77 / 153.86 / 0.460 | 43.99 / 0.197 | 19.96 / 116.28 / 0.564 | 0 |
| DDRM-100 | 28.84 / 40.52 / 0.214 | 26.47 / 45.22 / 0.273 | 25.88 / 0.156 | 36.17 / 15.32 / 0.119 | 100 |
| DPS | 24.71 / 34.69 / 0.304 | 22.38 / 41.01 / 0.348 | N/A | 24.89 / 32.64 / 0.288 | 250 |
| DDNM-100 | 28.85 / 35.13 / 0.206 | 26.53 / 44.22 / 0.272 | 23.65 / 0.138 | 38.70 / 4.48 / 0.062 | 100 |
| **SSD-100 (ours)** | 28.84 / 32.41 / 0.202 | 26.44 / 42.42 / 0.267 | 23.62 / 0.138 | 38.62 / 4.36 / 0.060 | 100 |
| DDRM-30 | 28.62 / 46.72 / 0.221 | 26.28 / 49.32 / 0.281 | 27.69 / 0.214 | 36.05 / 15.71 / 0.122 | 30 |
| DDNM-30 | 28.76 / 41.36 / 0.213 | 26.41 / 48.25 / 0.277 | 25.25 / 0.184 | 37.40 / 6.65 / 0.084 | 30 |
| **SSD-30 (ours)** | 28.71 / 36.77 / 0.208 | 26.32 / 44.97 / 0.271 | 24.11 / 0.159 | 38.34 / 4.98 / 0.065 | 30 |

| ImageNet | SR × 4 | SR × 8 | Colorization | Deblur (gauss) | NFEs↓ |
|---|---|---|---|---|---|
| Method | PSNR↑ / FID↓ / LPIPS↓ | PSNR↑ / FID↓ / LPIPS↓ | FID↓ / LPIPS↓ | PSNR↑ / FID↓ / LPIPS↓ | |
| $H^†y$ | 26.26 / 106.01 / 0.322 | 22.86 / 124.89 / 0.4690 | 27.40 / 0.231 | 19.33 / 102.33 / 0.553 | 0 |
| DDRM-100 | 27.40 / 43.27 / 0.260 | 23.74 / 83.08 / 0.420 | 36.44 / 0.224 | 36.48 / 11.81 / 0.121 | 100 |
| DPS | 20.34 / 72.33 / 0.485 | 18.38 / 76.89 / 0.538 | N/A | 24.89 / 32.64 / 0.288 | 250 |
| DDNM-100 | 27.44 / 39.42 / 0.251 | 23.80 / 80.09 / 0.412 | 36.46 / 0.219 | 40.48 / 3.33 / 0.041 | 100 |
| **SSD-100 (ours)** | 27.45 / 37.69 / 0.248 | 23.76 / 82.11 / 0.409 | 35.40 / 0.215 | 40.32 / 3.07 / 0.039 | 100 |
| DDRM-30 | 27.17 / 46.14 / 0.269 | 23.50 / 84.53 / 0.426 | 36.48 / 0.237 | 35.90 / 13.35 / 0.130 | 30 |
| DDNM-30 | 27.22 / 40.12 / 0.256 | 23.53 / 74.60 / 0.414 | 36.46 / 0.229 | 37.67 / 6.91 / 0.081 | 30 |
| **SSD-100 (ours)** | 27.13 / 38.24 / 0.251 | 23.44 / 76.35 / 0.411 | 36.22 / 0.223 | 39.23 / 4.64 / 0.053 | 30 |

Table 1: Quantitative evaluation on the **CelebA**(*top*) and **ImageNet**(*bottom*) datasets for various typical IR tasks. Red indicates the best performance.

Tumanyan et al., 2023) have indicated that diffusion models typically generate the overall layout and color in the early stage, generate the structure and appearance in the middle stage, and generate the texture details in the final stage. We notice that SSD employs shortcut sampling to skip the early stage and ensure the preservation of the overall layout. However, during the middle final stage, the utilization of back projection with an inaccurate $H$ has the potential to deteriorate the fine-textured details, leading to suboptimal outcomes. In $SSD^+$, rather than performing back projection throughout the generation process, we restrict its use to the middle stage, where it still plays a crucial role in maintaining structure consistency. During the final stage of generation, we rely exclusively on diffusion priors to ensure texture details without compromising the integrity of the original structure.

## 4 EXPERIMENTS

### 4.1 EXPERIMENTAL SETUP

**Pretrained Models and Datasets** To evaluate the performance of SSD, we conduct experiments on two datasets with different distribution characters: CelebA 256×256 (Karras et al., 2017) for face images and ImageNet 256×256 (Deng et al., 2009) for natural images, both containing 1k validation images independent of the training dataset. For CelebA 256×256, we use the denoising network VP-SDE(Song et al., 2021c; Meng et al., 2021), which is pre-trained by Meng et al. (2021)[1]. For ImageNet 256×256, we use the denoising network guided-diffusion (Dhariwal & Nichol, 2021), which is pre-trained by Dhariwal & Nichol (2021)[2].

**Degradation Operators** We conduct experiments on several typical IR tasks, including Super-Resolution($\times 4$, $\times 8$), Colorization, Inpainting and Deblurring. Details of degradation operators can be found in Appendix E.1

**Evaluation** We use PSNR, FID(Heusel et al., 2017), and LPIPS(Zhang et al., 2018b) as the main metrics to quantitatively evaluate the performance of image restoration. Especially due to

---

[1]Pre-trained Model files can be downloaded here provided by SDEdit

[2]Pre-trained Model files can be downloaded here provided by guided-diffusion

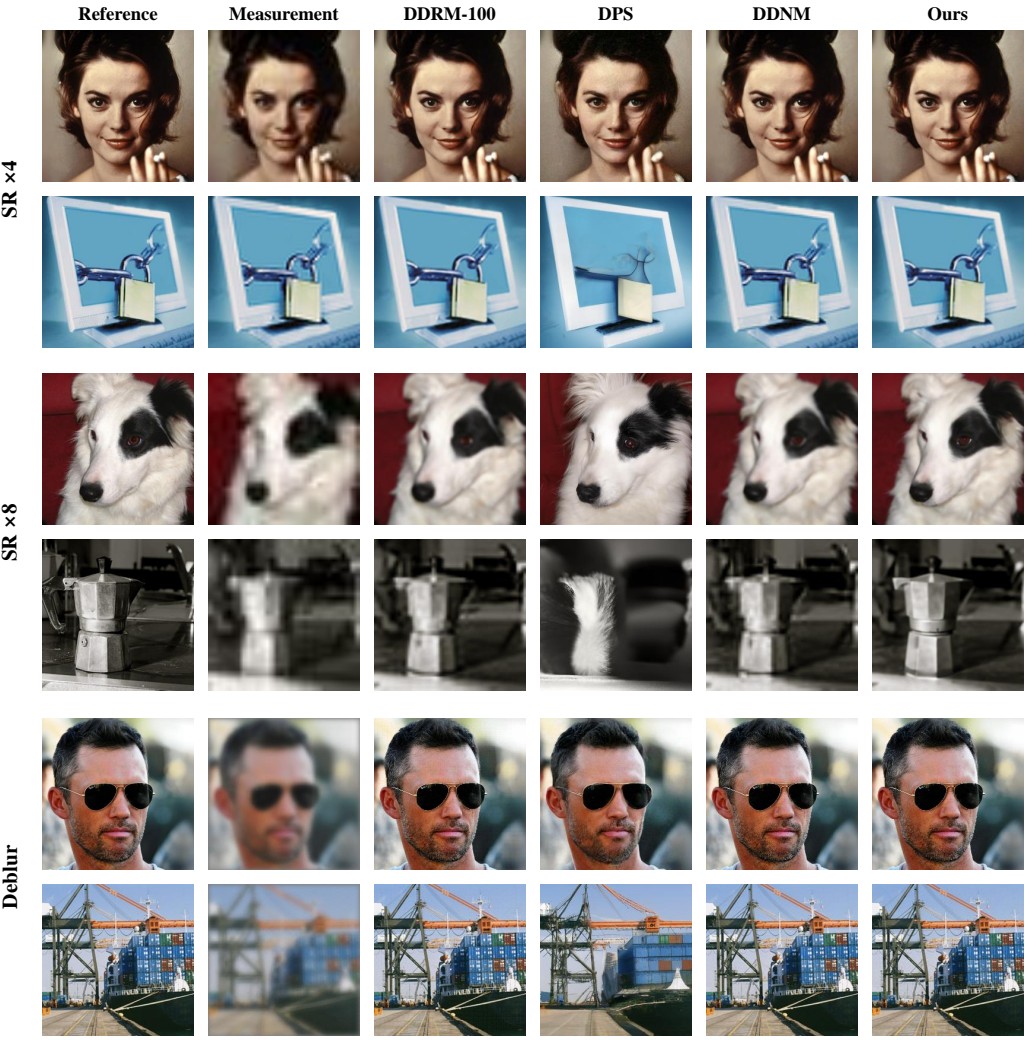

Figure 4: Qualitative results of different zero-shot IR methods on CelebA and Imagenet Dataset.

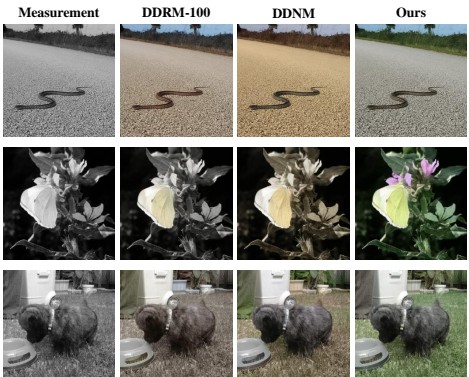

Figure 5: Colorization results of different zero-shot IR methods on ImageNet Dataset

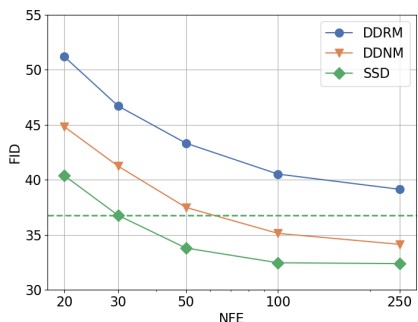

Figure 6: Comparison of the performance of various methods affected by NFEs with SR $\times$ 4 task on CelebA dataset

the inability of PSNR to capture colorization performance, we use FID and LPIPS for colorization. Additionally, we use Neural Function Evaluations(NFEs) as the metrics of sampling speed, which is a commonly employed benchmark in diffusion model-based methods. Since SSD introduces an additional inversion process, the NFEs of SSD are calculated by summing the steps involved in both the inversion process and the generation process.

**Comparison Methods**   We compare the restoration performance of the proposed method with recent State-Of-The-Art zero-shot image restoration methods using pre-trained diffusion models: DDRM(Kawar et al., 2022), DPS(Chung et al., 2023) and DDNM(Wang et al., 2023). For a fair comparison, all methods above use the same pre-trained denoising networks and degradation operator.

## 4.2 Noiseless Image Restoration Results

We compare SSD (with 30 and 100 steps) with previous methods mentioned in Sec 4.1. The quantitative evaluation results shown in Table 1 illustrate that the proposed method achieves competitive results compared to state-of-the-art methods. When setting NFE to 30, SSD-30 outperforms other methods known for fast sampling like DDRM-30 and achieves better perception-oriented metris (i.e., FID, LPIPS) than SOTA methods (DDNM with 100 NFEs). When setting NFE to 100, SSD-100 achieves SOTA performance in many IR tasks, including SR $\times 4$ and colorization. As shown in Fig. 4, 5, SSD generates high-quality restoration results in all tested datasets and tasks.

We further explore the performance of various methods in terms of FID with respect to the change in NFEs, which is shown in Fig. 6. We conduct experiments with SR $\times 4$ task on the CelebA dataset. Results show that SSD outperforms all the other methods in both high NFEs.

## 4.3 Noisy Image Restoration Results

To illustrate the robustness of $SSD^+$ in the face of noisy situations and complex degradation, we evaluate SSD and $SSD^+$ on diverse inverse problems with gaussian noise and JPEG compression(Shin & Song, 2017). For gaussian noise, we add gaussian noise $z \sim \mathcal{N}(0, \sigma^2)$ to the degraded image $y$, where $\sigma$ represents the intensity of noise and is randomly distributed in $[0.0, 0.2]$. For JPEG compression, we perform JPEG compression(Wang et al., 2021) with a quality factor of 60 after the degraded operator $H$ is applied. The quantitative result is shown in Tab. 2. Qualitative results are available in Fig. 7

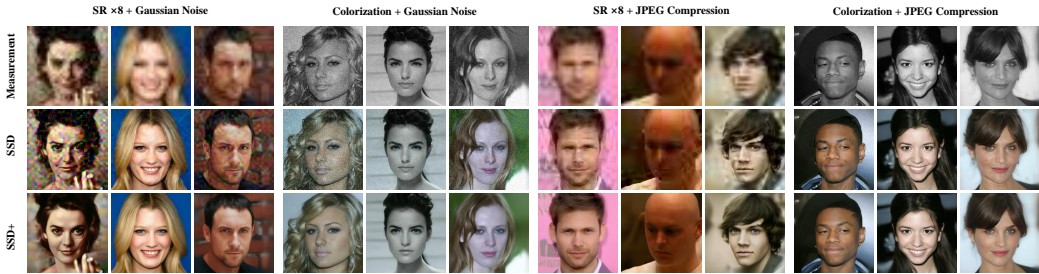

Figure 7: Qualitative results on CelebA of solving inverse problems with additional Gaussian noise and JEPG compression.

| CelebA | 8× SR + Noise | Colorization + Noise | 8× SR + JPEG | Colorization + JPEG |
|---|---|---|---|---|
| Method | PSNR↑ / LPIPS↓ / FID↓ | LPIPS↓ / FID↓ | PSNR↑ / LPIPS↓ / FID↓ | LPIPS↓ / FID↓ |
| SSD | 22.51 / 0.528 / 85.92 | 0.533 / 68.24 | 23.23 / 0.414 / 80.74 | 0.301 / 48.54 |
| $SSD^+$ | 24.60 / 0.299 / 43.84 | 0.373 / 45.02 | 24.23 / 0.301 / 46.32 | 0.372 / 45.02 |

Table 2: Quantitative evaluation on CelebA of solving inverse problems with additional **Gaussian noise**(*left*) and **JEPG compression**(*right*). Red indicates the best performance.

## 5 Conclusion

In this paper, we propose SSD, a novel framework for solving inverse problems in a zero-shot manner. We have departed from the conventional "Noise-Target" paradigm and instead proposed a shortcut sampling pathway of "Input-Embryo-Target". This novel approach enables us to achieve satisfactory results with reduced steps. We further propose $SSD^+$, an enhanced version of SSD tailored to excel in scenarios where degradation estimation is less accurate or in the presence of noise. We hope the proposed pipeline may inspire future work on inverse problems to solve them in a more efficient manner.

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

# A  PROOFS

**Definition 1.** *We define the iterative process of Distortion Adaptive Inversion as:*

$$x_{t+1} = \sqrt{\alpha_{t+1}} f_\theta(x_t, t) + \sqrt{1 - \alpha_{t+1} - \eta\beta_{t+1}} \epsilon_\theta(x_t, t) + \sqrt{\eta\beta_{t+1}} z \tag{11}$$

*where $\eta$ control the proportion of random disturbances and $0 < \eta < 1$.*

*For ease of exposition, the predicted noise in DA Inversion at each time-step can be rephrased as:*

$$\epsilon_{DA} = \frac{1}{\sqrt{1 - \alpha_{t+1}}} (\sqrt{1 - \alpha_{t+1} - \eta\beta_{t+1}} \epsilon_\theta(x_t, t) + \sqrt{\eta\beta_{t+1}} z), z \sim \mathcal{N}(\mu, \sigma) \tag{12}$$

**Proposition 1.** *Assuming $\epsilon_\theta(x_t, t) \sim \mathcal{N}(\mu, \sigma^2)$, We have:*

$$\epsilon_{DA} \sim \mathcal{N}(\frac{\sqrt{1 - \alpha_{t+1} - \eta\beta_{t+1}}}{\sqrt{1 - \alpha_{t+1}}} \mu, 1 + \frac{1 - \alpha_{t+1} - \eta\beta_{t+1}}{1 - \alpha_{t+1}} (\sigma^2 - 1)) \tag{13}$$

*thus:*

$$\begin{aligned} \|\mu_{\epsilon_{DA}}\| &< \|\mu\| \\ \|\sigma_{\epsilon_{DA}}^2 - 1\| &< \|\sigma^2 - 1\| \end{aligned} \tag{14}$$

*which indicates that after adding random disturbance, $\epsilon_{DA}$ becomes closer to $\mathcal{N}(0, 1)$.*

*Proof.* According to Eq. 12, we can rewrite it based on reparameterization techniques:

$$z_1 = \frac{\sqrt{1 - \alpha_{t+1} - \eta\beta_{t+1}}}{\sqrt{1 - \alpha_{t+1}}} \epsilon_\theta(x_t, t) \tag{18}$$

$$z_2 = \frac{\sqrt{\eta\beta_{t+1}}}{\sqrt{1 - \alpha_{t+1}}} z \tag{19}$$

$$\epsilon_{DA} = z_1 + z_2 \tag{20}$$

further:

$$z_1 \sim \mathcal{N}(\frac{\sqrt{1 - \alpha_{t+1} - \eta\beta_{t+1}}}{\sqrt{1 - \alpha_{t+1}}} \mu, \frac{1 - \alpha_{t+1} - \eta\beta_{t+1}}{1 - \alpha_{t+1}} \sigma^2) \tag{21}$$

$$z_2 \sim \mathcal{N}(0, \frac{\eta\beta_{t+1}}{1 - \alpha_{t+1}}) \tag{22}$$

Since $z$ is sampled randomly and independently from the standard normal distribution, it is mutually independent of $\epsilon_\theta(x_t, t)$. Additionally, $z_1$ and $z_2$ are also mutually independent. Therefore we have:

$$\mu_{\epsilon_{DA}} = \mu_{z_1} + \mu_{z_2} \tag{23}$$

$$= \frac{\sqrt{1 - \alpha_{t+1} - \eta\beta_{t+1}}}{\sqrt{1 - \alpha_{t+1}}} \mu \tag{24}$$

$$\sigma_{\epsilon_{DA}}^2 = \sigma_{z_1}^2 + \sigma_{z_2}^2 \tag{25}$$

$$= \frac{1 - \alpha_{t+1} - \eta\beta_{t+1}}{1 - \alpha_{t+1}} \sigma^2 + \frac{\eta\beta_{t+1}}{1 - \alpha_{t+1}} \tag{26}$$

$$= 1 + \frac{1 - \alpha_{t+1} - \eta\beta_{t+1}}{1 - \alpha_{t+1}} (\sigma^2 - 1) \tag{27}$$

thus we have:

$$0 < \|\mu_{\epsilon_{DA}}\| < \|\mu\| \tag{28}$$

$$\begin{cases} 1 < \sigma^2_{\epsilon_{DA}} < \sigma^2, \sigma^2 > 1 \\ \sigma^2 < \sigma^2_{\epsilon_{DA}} < 1, \sigma^2 < 1 \end{cases} \tag{29}$$

which indicates that by introducing random disturbance, the mean of $\epsilon_{DA}$ tends to approach $0$ in comparison to the original noise mean $\mu$, while the variance of $\epsilon_{DA}$ tends to approach $1$ in comparison to the original noise variance $\sigma^2$. As a result, $\epsilon_{DA}$ exhibits a closer resemblance to the standard normal distribution $z \sim \mathcal{N}(0, 1)$.

□

# B  THE DEVIATION FROM STANDARD NORMAL DISTRIBUTION DURING THE INVERSION PROCESS

We observe that the predicted noise $\epsilon_\theta(x_t, t)$ during DDIM Inversion process deviates from the standard normal distribution. As the diffusion models are pretrained to generate high-quality images from the standard normal distribution, employing out-of-domain input distributions may undermine the generation performance, leading to unrealistic results. We further define this as the following assumption:

**Assumption 1.** *Diffusion Models rely on in-domain noise distribution to generate high-quality images. When facing low-quality input images, the distribution of predicted noise $\epsilon_\theta(x_t, t)$ during the DDIM Inversion process exhibits a greater deviation from the standard normal distribution.*

To provide further illustration, we preformed validation experiments on ImageNet dataset. We randomly select 100 images from ImageNet validation dataset and apply DDIM Inversion on input images with varying degrees of degradation, including original high-quality images, as well as images downscaled by factors of 2, 4 and 8 using bicubic interpolation. We utlize the Kullback-Leibler (KL) divergence as a metric to quantify the deviation from the standard normal distribution, given by:

$$D_{KL}(P\|Q) = \int_{-\infty}^{\infty} p(x) \log\left(\frac{p(x)}{q(x)}\right) dx \tag{30}$$

As depicted in Fig. 8, it can be observed that as the downsampling scale increases, the noise in the DDIM inversion process deviates further from the standard normal distribution. In contrast, during the proposed DA Inversion process, the predicted noise approaches the standard normal distribution $z \sim \mathcal{N}(0, 1)$. This outcome validates our previous hypothesis stated in Prop. 1.

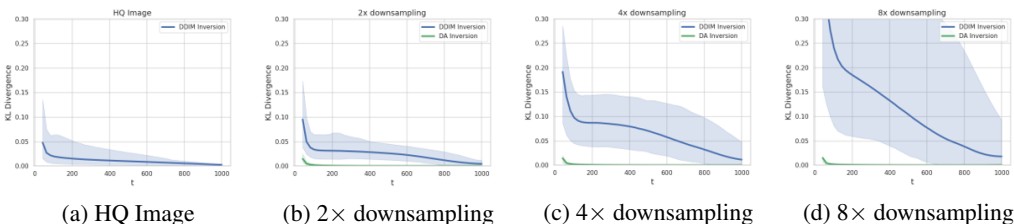

| (a) HQ Image | (b) $2\times$ downsampling | (c) $4\times$ downsampling | (d) $8\times$ downsampling |

Figure 8: Deviation of noise during the inversion process of differently degraded images. We observe that with the increase in the degree of image degradation, the deviation between the predicted noise and the standard normal distribution also increases. In contrast, the proposed DA Inversion does not exhibit such deviation.

## C  CONNECTIONS BETWEEN DDIM INVERSION, DDPM INVERSION, AND DISTORTION ADAPTIVE INVERSION

In the main text, we attribute the failure of DDIM Inversion to the deviation from the standard normal distributions. In this section, we further explore the reasons behind the out-of-domain results produced by DDIM Inversion after multiple iterations in the inversion process.

Let's first review the forward process in diffusion models, which can be expressed as follows(Ho et al., 2020):

$$x_{t+1} = \sqrt{\alpha_{t+1}}x_0 + \sqrt{1 - \alpha_{t+1}}\bar{\epsilon}_{t+1}, \bar{\epsilon}_{t+1} \sim \mathcal{N}(0, 1) \tag{1}$$

$$x_{t+1} = \sqrt{1 - \beta_{t+1}}x_t + \sqrt{\beta_{t+1}}\epsilon, \epsilon \sim \mathcal{N}(0, 1) \tag{2}$$

And DDIM Inversion process is defined as:

$$x_{t+1} = \sqrt{\alpha_{t+1}}f_\theta(x_t, t) + \sqrt{1 - \alpha_{t+1}}\epsilon_\theta(x_t, t) \tag{9}$$

Notice that the forward process in Eq. 1 and the DDIM Inversion process in Eq. 9 share a similar form, wherein $f_\theta(x_t, t)$ substitutes $x_0$ and $\epsilon_\theta(x_t, t)$ replaces the noise $\bar{\epsilon}_{t+1}$. It is important to note that $\epsilon_\theta(x_t, t)$ is the approximation of the noise added until time-step $t$, and $\bar{\epsilon}_{t+1}$ represents the noise added until time-step $t + 1$. To account for this misalignment, we rewrite Eq. 1 as follows:

$$
\begin{aligned}
x_{t+1} &= \sqrt{1 - \beta_{t+1}}x_t + \sqrt{\beta_{t+1}}\epsilon_{t+1} \\
&= \sqrt{(1 - \beta_{t+1})(1 - \beta_t)}x_{t-1} + \sqrt{(1 - \beta_{t+1})\beta_t}\epsilon_t + \sqrt{\beta_{t+1}}\epsilon_{t+1} \\
&= \dots \\
&= \sqrt{\alpha_t}x_0 + \sqrt{1 - \alpha_{t+1}}\bar{\epsilon}_{t+1}
\end{aligned}
\tag{31}
$$

where:

$$
\begin{aligned}
\sqrt{1 - \alpha_{t+1}}\bar{\epsilon}_{t+1} &= \sqrt{\beta_{t+1}}\epsilon_{t+1} + \sqrt{(1 - \beta_{t+1})\beta_t}\epsilon_t + \sqrt{(1 - \beta_{t+1})(1 - \beta_t)\beta_{t-1}}\epsilon_{t-1} \\
&\quad + \dots + \sqrt{(1 - \beta_{t+1})(1 - \beta_t)\dots(1 - \beta_2)\beta_1}\epsilon_1 \\
&= \sqrt{\beta_{t+1}}\epsilon_{t+1} + \sqrt{(1 - \beta_{t+1})\beta_t + (1 - \beta_{t+1})(1 - \beta_t)\beta_{t-1} + \dots}\bar{\epsilon}_t \\
&= \sqrt{\beta_{t+1}}\epsilon_{t+1} + \sqrt{(1 - \alpha_{t+1}) - \beta_{t+1}}\bar{\epsilon}_t
\end{aligned}
\tag{32}
$$

therefore we have:

$$x_{t+1} = \sqrt{\alpha_{t+1}}x_0 + \underbrace{\sqrt{(1 - \alpha_{t+1}) - \beta_{t+1}}\bar{\epsilon}_t}_{\text{Synthesis of the first }t\text{-term noise}} + \underbrace{\sqrt{\beta_{t+1}}\epsilon_{t+1}}_{\text{the }t + 1 \text{ term noise}} \tag{33}$$

where $\epsilon_{t+1}$ is independent of $\bar{\epsilon}_t$.

During the forward (inversion) process from $t$ to $t + 1$, we can interpret $\bar{\epsilon}_t$ as the existing information in $x_t$, and $\epsilon_{t+1}$ as the additional information. By setting $\bar{\epsilon}_t = \epsilon_\theta(x_t, t) \sim \mathcal{N}(0, 1)$ and maintain $\epsilon_{t+1}$ stochastic (where $\epsilon_\theta(x_t, t)$ is independent of $\epsilon_{t+1}$ ), we can build a fully stochastic process. This is equivalent to the forward process described in Eq. 2, which is referred to as "DDPM Inversion":

$$
\begin{aligned}
x_{t+1} &= \sqrt{\alpha_{t+1}} f_\theta(x_t, t) + \sqrt{(1 - \alpha_{t+1}) - \beta_{t+1}} \epsilon_\theta(x_t, t) + \sqrt{\beta_{t+1}} z \\
&= \sqrt{\alpha_{t+1}} \frac{x_t - \sqrt{1 - \alpha_t} \epsilon_\theta(x_t, t)}{\sqrt{\alpha_t}} + \sqrt{(1 - \alpha_{t+1}) - \beta_{t+1}} \epsilon_\theta(x_t, t) + \sqrt{\beta_{t+1}} z \\
&= \sqrt{\alpha_{t+1}} x_t - \sqrt{\frac{(1 - \alpha_t) \alpha_{t+1}}{\alpha_t}} + \sqrt{1 - \alpha_{t+1} - \beta_{t+1}} \epsilon_\theta(x_t, t) + \sqrt{\beta_{t+1}} z \\
&= \sqrt{\alpha_{t+1}} x_t - \sqrt{\frac{\alpha_{t+1}}{\alpha_t} - \alpha_{t+1}} \epsilon_\theta(x_t, t) + \sqrt{1 - \alpha_{t+1} - \beta_{t+1}} \epsilon_\theta(x_t, t) + \sqrt{\beta_{t+1}} z \\
&= \sqrt{\alpha_{t+1}} x_t - \sqrt{1 - \alpha_{t+1} - \beta_{t+1}} \epsilon_\theta(x_t, t) + \sqrt{1 - \alpha_{t+1} - \beta_{t+1}} \epsilon_\theta(x_t, t) + \sqrt{\beta_{t+1}} z \\
&= \sqrt{\alpha_{t+1}} x_t + \sqrt{\beta_{t+1}} z
\end{aligned}
\tag{34}
$$

Similarly, we can rewrite Eq. 9 as a similar form to Eq. 33, given by:

$$
\begin{aligned}
x_{t+1} &= \sqrt{\alpha_{t+1}} x_0 + \underbrace{\sqrt{1 - \alpha_{t+1} - \beta_{t+1}} \epsilon_\theta(x_t, t)}_{\text{Synthesis of the first } t\text{-term noise}} + \underbrace{(\sqrt{1 - \alpha_{t+1}} - \sqrt{1 - \alpha_{t+1} - \beta_{t+1}}) \epsilon_\theta(x_t, t)}_{\text{the } t + 1 \text{ term noise}} \\
&= \sqrt{\alpha_{t+1}} x_t + (\sqrt{1 - \alpha_{t+1}} - \sqrt{1 - \alpha_{t+1} - \beta_{t+1}}) \epsilon_\theta(x_t, t)
\end{aligned}
\tag{35}
$$

We adjust the coefficient of the $t + 1$ term noise because they do not satisfy independence. DDIM Inversion is a deterministic process because the $t + 1$ term noise inherits the information of $x_t$.

Further, the proposed DA Inversion takes into account both the information inherited from $x_t$ and additional randomness. The formulation is as follows.

$$
\begin{aligned}
x_{t+1} &= \sqrt{\alpha_{t+1}} x_0 + \underbrace{\sqrt{1 - \alpha_{t+1} - \beta_{t+1}} \epsilon_\theta(x_t, t)}_{\text{Synthesis of the first } t\text{-term noise}} + \\
&\quad \underbrace{(\sqrt{1 - \alpha_{t+1} - \eta\beta_{t+1}} - \sqrt{1 - \alpha_{t+1} - \beta_{t+1}}) \epsilon_\theta(x_t, t) + \sqrt{\eta\beta_{t+1}} z}_{\text{the } t + 1 \text{ term noise}} \\
&= \sqrt{\alpha_{t+1}} x_t + (\sqrt{1 - \alpha_{t+1} - \eta\beta_{t+1}} - \sqrt{1 - \alpha_{t+1} - \beta_{t+1}}) \epsilon_\theta(x_t, t) + \sqrt{\eta\beta_{t+1}} z
\end{aligned}
\tag{36}
$$

The comparison of different inversion methods, namely DDIM Inversion (Song et al., 2021a), DDPM Inversion (Ho et al., 2020), and our proposed Distorted Adaptive Inversion, is summarized in Table 3. We also visualize the inversion process of different methods, as shown in Fig. 9 10.

| | | DDIM Inversion | DDPM Inversion | DA Inversion (ours) |
|---|---|---|---|---|
| **Inversion** | | | | |
| Inversion Process | | Equation 9 | Equation 33 | Equation 11 |
| **Composition of $t + 1$ term noise** | | | | |
| Coefficient of $\epsilon_\theta(x_t, t)$ | $c(\epsilon_\theta)$ | $\sqrt{1 - \alpha_{t+1}}$ $- \sqrt{1 - \alpha_{t+1} - \beta_{t+1}}$ | $0$ | $\sqrt{1 - \alpha_{t+1} - \eta\beta_{t+1}}$ $- \sqrt{1 - \alpha_{t+1} - \beta_{t+1}}$ |
| Coefficient of $z$ | $c(z)$ | $0$ | $\sqrt{\beta_{t+1}}$ | $\sqrt{\eta\beta_{t+1}}$ |
| **Performance** | | | | |
| Faithfulness | | ✓ | ✗ | ✓ |
| Realism | | ✗ | ✓ | ✓ |

Table 3: Comparisons on different inversion methods.

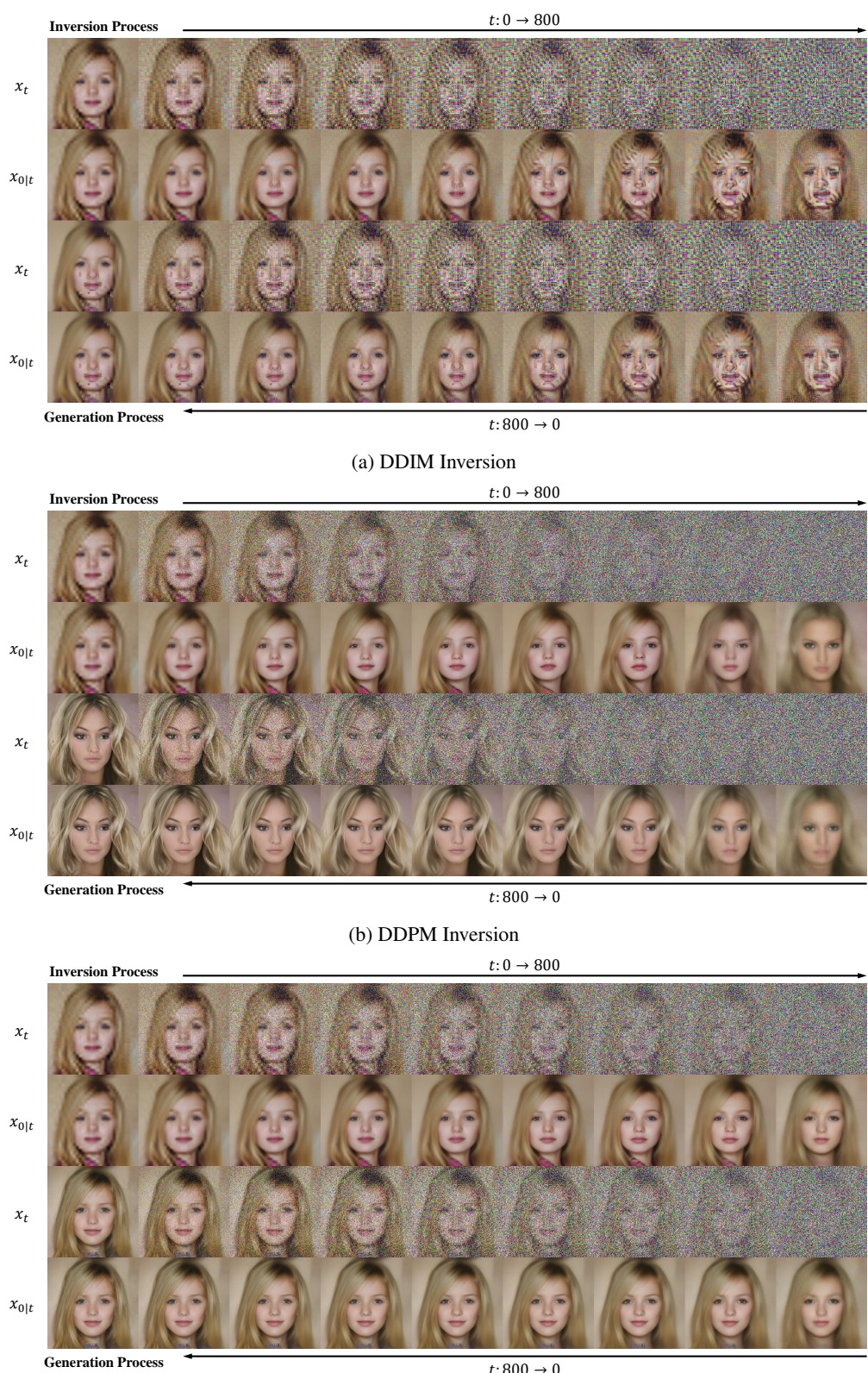

Figure 9: Visualization results of various inversion process on CelebA

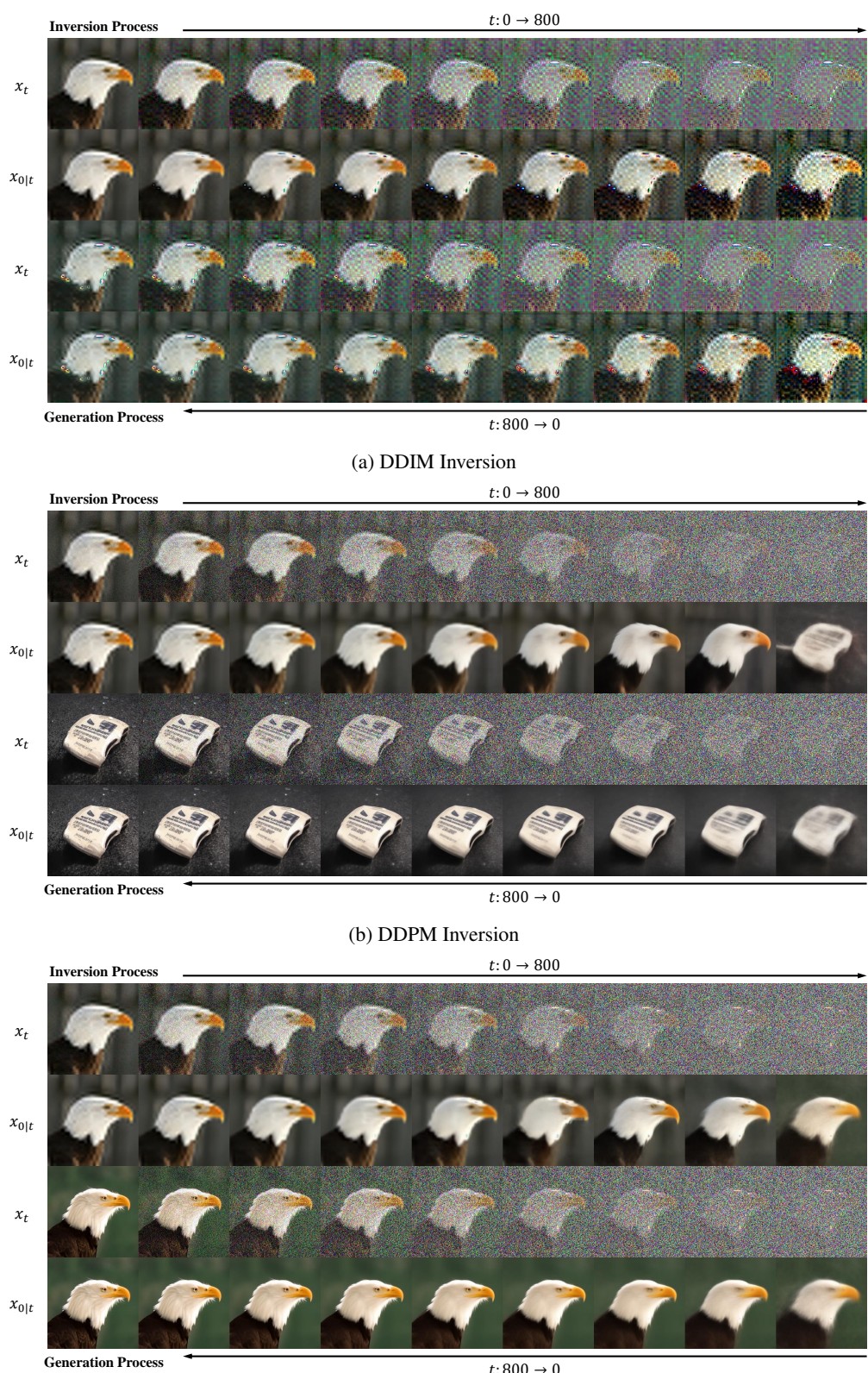

Figure 10: Visualization results of various inversion process on Imagenet

# D  COMPARISON BETWEEN SSD AND DDNM

In this section, we provide a detailed comparison between proposed SSD and DDNM(Wang et al., 2023). The key idea of DDNM is "null space decomposition". They decompose the restored image as a null-space part $(I - H^\dagger H)x$ and a range-space part $+H^\dagger y$. The former represents the identity information of the input LQ image, and the latter represents the detailed information of the HQ image. During the generation process, DDNM keeps the range-space part unchanged to force consistency and obtain the null-space part through iterative refinement. The back projection in SSD actually shares a similar form of the "null space decomposition" in DDNM. The main differences between SSD and DDNM are the following aspects:

**Key ideas**  The key idea of DDNM revovles around "null space decomposition". They rederive the back projection through range-null space decomposition and propose manually designed degenerate operators to solve common inverse problems. Their primary emphasis lies in substantiating the efficacy of "null space decomposition" through both theoretical derivations and empirical experiments.

Conversely, **our** key idea is to find the "embryo", a transitional state that bridges the input low-quality image and the desired high-quality results. The "Embryo" holds the potential to accelerate the whole process. So we mainly focus on how to improve the inversion process to obtain an "Embryo" that both satisfies faithfulness and realism. Back projection is regarded as a technique for improving consistency in our approach.

**Pipeline:**  DDNM considers solving the inverse problem in a generation pipeline of "noise-Target". They start from the random noise and adopt additional consistency conditions to generate the expected results. However, this methodology entails the inclusion of numerous superfluous steps, which are expended in recreating the layout and structure that are inherently present in the input image.

Instead, we propose a novel pipeline of "Input - Embryo - Target". We utilize DA Inversion to effectively obtain a semi-manufactured restoration result named "embryo", and start the generation process from the "embryo" instead of random noise. Through the proposed shortcut pipeline, we enable skipping the early stage of generation and preserving the layout information of the input image with only a minimal number of inversion steps, allowing the entire process to proceed in reduced steps.

**Solving noisy situations:**  DDNM supports noisy restoration by treating the noise of the degradation process as an integral component of the forward process noise within the diffusion framework. This approach incorporates a denoising step within the diffusion process, thereby extending its applicability to tasks involving noisy inputs. The enhanced version, DDNM+, excels in managing highly noisy scenarios but necessitates the incorporation of additional denoising steps ("time-travel") and carefully selecting noise parameters for each specific case.

In our work, we find the failure of noisy situations in SSD mainly attributed to, the constraint invalidation of back projection, when faced with inaccurate estimation of $H$. Given that our method does not overly depend on back projection, we are able to handle noisy scenarios by simply omitting the back projection during the final generation phase. With SSD+, we exhibit the capability to address additional noise and complex degradation within a certain range, all without the need for additional parameter adjustments.

# E  EXPERIMENTAL DETAILS

## E.1  DEGRADATION OPERATORS

(i) For super-resolution, we use a bicubic filter to downsample the image at a given specific scale. (ii) For colorization, we calculate the average value of the red, green and blue channels of the original image to obtain the grayscale image. (iii) For deblurring, we select a Gaussian blur kernel with size $9 \times 9$ as the blur kernel, and the width is set to $\sigma = 15.0$. (iv) For inpainting(random) task, we randomly drop the pixels in the image with a probability of 0.5; (iv) For inpainting(box) task, we randomly mask images with a 96x96 rectangle.

Following DDNM(Wang et al., 2023), we utilize manually designed degradation operators $H$ and corresponding $H^\dagger$ in our methods. Details about operators are listed as follows:

**Super Resolution**  For super resolution, we use the bicubic downsampling kernel as the degradation operator $H$. The weight formula of bicubic interpolation is:

$$
w(t) = \begin{cases}
\frac{1}{6}\left((t+2)^3 - 4(t+1)^3 + 6t^3\right), & \text{if } |t| \leq 1 \\
-\frac{1}{6}\left(t^3 - 4(t-1)^3 + 6(t-2)^3\right), & \text{if } 1 < |t| \leq 2 \\
0, & \text{otherwise}
\end{cases}
\tag{37}
$$

We calculate the pseudo-inverse $H^\dagger$ using singular value decomposition(SVD)(Kawar et al., 2022), which can be expressed as follows:

$$
H = U\Sigma V^T \tag{38}
$$

where $U$ and $V$ are orthogonal matrices, $\Sigma$ is a diagonal matrix, and the elements on $\Sigma$ are the singular values of H. Then we can obtain the pseudo-inverse $H^\dagger$ through:

$$
H^\dagger = V\Sigma^\dagger U^T \tag{39}
$$

$\Sigma^\dagger$ is the pseudo-inverse of $\Sigma$, which can be calculated by taking the reciprocal of non-zero elements on the diagonal of $\Sigma$, while keeping the zero elements unchanged.

**Colorization**  For colorization, by choosing $H = [1/3, 1/3, 1/3]$ as a pixel-wise degradation operator, we can convert each pixel from the rgb value $[r, g, b]^T$ into the grayscale value $[\frac{r}{3} + \frac{g}{3} + \frac{b}{3}]$. And the pseudo-inverse is $H^\dagger = [1, 1, 1]^T$, which satisfies $HH^\dagger = I$.

**Deblurrring**  For deblurring, we use the gaussian blur kernel as the degradation operator. The gaussian blur kernel can be expressed as:

$$
w(i, j) = \frac{1}{2\pi\sigma^2} e^{-\frac{i^2+j^2}{2\sigma^2}} \tag{40}
$$

We set the width as $\sigma = 15.0$ and the kernel size as $9 \times 9$. Similar to super resolution, we use SVD to obtain the pseudo-inverse $H^\dagger$.

### E.2 REPRODUCIBILITY DETAILS

**Hyperparameter Settings**  Here, we provide the detailed hyperparameter settings of SSD for various inverse problems.

- SSD of 100 NFEs
    - Super-Resolution, Inpainting and Deblurring:
        * Inversion step: $S_{inv} = 15$;
        * Generation step: $S_{gen} = 85$;
        * Shortcut time-step: $t_0 = 550$;
        * Proportion of added disturbance: $\eta = 0.4$
    - Colorization
        * inversion step: $S_{inv} = 15$;
        * generation step: $S_{gen} = 85$;
        * Shortcut time-step: $t_0 = 750$;
        * Proportion of added disturbance: $\eta = 0.8$
- SSD of 30 NFEs
    - Super-Resolution, Inpainting and Deblurring:
        * inversion step: $S_{inv} = 5$;
        * generation step: $S_{gen} = 25$;
        * Shortcut time-step: $t_0 = 550$;

   * Proportion of added disturbance: $\eta = 0.4$
  – Colorization
   * inversion step: $S_{inv} = 5$;
   * generation step: $S_{gen} = 25$;
   * Shortcut time-step: $t_0 = 750$;
   * Proportion of added disturbance: $\eta = 0.8$

**Pretrained models** For CelebA 256×256, we use the denoising network VP-SDE(Song et al., 2021c; Meng et al., 2021), which is pre-trained on CelebA. Pre-trained Model files can be downloaded here provided by SDEdit.

For ImageNet 256×256, we use the denoising network guided-diffusion (Dhariwal & Nichol, 2021), which is pre-trained on ImageNet. Pre-trained Model files can be downloaded here provided by guided-diffusion.

**Code Availability** We will open-source our code of SSD upon publication to enhance reproducibility.

### E.3 ALGORITHM

---
**Algorithm 1** Image Restoration of SSD

---
**Input:** input measurement image $y$, degradation operator $H$
**Parameters:** inversion step $S_{inv}$, generation step $S_{gen}$, shortcut time-step $t_0$, proportion of added disturbance $\eta$
**Output:** restored Image $x$

  Step 1: Get the Embryo through DA Inversion
1: $x_0 = H^\dagger y$
2: Define $\{\tau_s\}_{s=1}^{s_{inv}}$, s.t. $\tau_1 = 0$, $\tau_{s_{inv}} = t_0$
3:
4: **for** $s = 1, \ldots, S_{inv} - 1$ **do**
5:  $\epsilon \leftarrow \epsilon_\theta(x_{\tau_s}, \tau_s, \varnothing)$;
6:  $x_{0|\tau_s} \leftarrow f_\theta(x_{\tau_s}, \tau_s)$;
7:  $z \sim \mathcal{N}(0,1)$
8:  $x_{\tau_{s+1}} = \sqrt{\alpha_{\tau_{s+1}}}x_{0|\tau_s} + \sqrt{1 - \alpha_{\tau_{s+1}} - \eta\beta_{\tau_{s+1}}}\epsilon + \sqrt{\eta\beta_{\tau_{s+1}}}z$   ▷ DA Inversion
9: **end for**
10:

  Step 2: Get HQ Image from the Embryo
11: Define $\{\tau_s\}_{s=1}^{s_{gen}}$, s.t. $\tau_1 = 0$, $\tau_{s_{gen}} = t_0$
12: **for** $s = S_{gen}, \ldots, 2$ **do**
13:  $\epsilon \leftarrow \epsilon_\theta(x_{\tau_s}, \tau_s, \varnothing)$;
14:  $x_{0|\tau_s} \leftarrow f_\theta(x_{\tau_s}, \tau_s)$    ▷ Denoising Step
15:  $\hat{x}_{0|\tau_s} \leftarrow (I - H^\dagger H)x_{0|\tau_s} + H^\dagger y$    ▷ Back Projection Step
16:  $x_{\tau_{s-1}} = \sqrt{\alpha_{\tau_{s-1}}}x_{0|\tau_s} + \sqrt{1 - \alpha_{\tau_{s-1}} - \sigma^2}\epsilon + \sigma z$
17: **end for**
18: **return** $x_0$

---

## F TIME CONSUMPTION

In our main paper, we use neural function evaluations(NFEs) as the metrics of sampling speed. Here we further measure the time consumption of different methods when restoring a single image. The results are illustrated in Tab. 4. We conduct super-resolution 4× experiments using a single Nvidia RTX 3090 GPU.

It is observerd that the projection-based methods, including DDRM(Kawar et al., 2022), DDNM(Wang et al., 2023), and our SSD, exhibit comparable time consumption(0.06 s/it for celeba, 0.11 s/it for

imagenet). The discrepancies in consumption mainly stem from variations in the implementation details of their code. On the other hand, the gradient-based methods such as DPS(Chung et al., 2023) require additional time and memory consumption, primarily due to the inclusion of back-propagation operations for gradient calculations at each step.

| Method | Celeba | | Imagenet | |
|---|---|---|---|---|
| | NFEs | Time(s/image) | NFEs | Time(s/image) |
| DDRM (Kawar et al., 2022) | 100 | 5.14 | 100 | 10.34 |
| DPS (Chung et al., 2023) | 100 | 12.70 | 100 | 36.87 |
| DDNM (Wang et al., 2023) | 100 | 4.98 | 100 | 10.11 |
| SSD(ours) | 100 | 5.03 | 100 | 10.27 |
| DDRM (Kawar et al., 2022) | 30 | 1.57 | 30 | 3.02 |
| DDNM (Wang et al., 2023) | 30 | 1.43 | 30 | 2.87 |
| SSD(ours) | 30 | 1.46 | 30 | 2.93 |

Table 4: Comparisons on Time Consumption of different methods.

# G ABLATION STUDIES

## G.1 DISTORATION ADAPTIVE INVERSION

We investigate the performance of the proposed DA Inversion by exploring the effects of different parameters. While SSD achieves competitive results under various parameters, the generation results of DA Inversion **alone** exhibit a noticeable trade-off between realism and faithfulness when choosing different values of $(t_0, \eta)$.

To further illustrate it, we randomly select 100 images from the Celeba-Test Dataset and apply 4× average-pooling downsampler on them as the input LQ Images. We then perform DA inversion with various $(t_0, \eta)$ values and proceed with the generation process. For *faithfulness*, we compute the average $L_2$ distance between the generation reuslts and the input LQ Images. For *realism*, we use FID to quantify the distribution difference. Fig. 11 illustrates the trade-off where faithfulness decreases and realism increases as $\eta$ increases. Notably, this trade-off follows an upward convex curve in Fig. 11c, suggesting that we can achieve a combination of faithfulness and realism by appropriately selecting hyperparameters.

## G.2 SSD

Here we extend our discussion to explore the impact of $(t_0, \eta)$ on the performance of SSD (DA Inversion + consistency constraints). To identify the most suitable combination, we perform experiments with different $(t_0, \eta)$ values for the super resolution 4× task on Celeba datasets. Results are presented in Tab. 5.

| $\eta$ | $t_0$ | PSNR (↑) | FID (↓) | LPIPS (↓) |
|---|---|---|---|---|
| 0.1 | 550 | 27.12 | 39.12 | 0.253 |
| 0.4 | 550 | 27.13 | 38.24 | 0.252 |
| 0.7 | 550 | 27.15 | 38.63 | 0.252 |
| 1.0 | 550 | 27.15 | 38.80 | 0.252 |

| $\eta$ | $t_0$ | PSNR (↑) | FID (↓) | LPIPS (↓) |
|---|---|---|---|---|
| 0.4 | 450 | 27.17 | 39.35 | 0.253 |
| 0.4 | 550 | 27.13 | 38.24 | 0.252 |
| 0.4 | 650 | 27.10 | 38.70 | 0.252 |
| 0.4 | 800 | 27.06 | 38.90 | 0.253 |

(a) Performance with different $\eta$.          (b) Performance with different $t_0$.

Table 5: Ablation studies on $\eta$ and $t_0$.(Applying both DA Inversion and back-projection)

Furthermore, we perform ablation studies to validate the effectiveness of the proposed modules in SSD. As depicted in Fig. 12 and Tab. 6, applying DDIM Inversion alone leads to faithful but unrealistic results, whereas applying DA Inversion alone produces faithful and realistic results. Moreover, the integration of consistency constraints in SSD, including back projection and attention injection, further improves the performance.

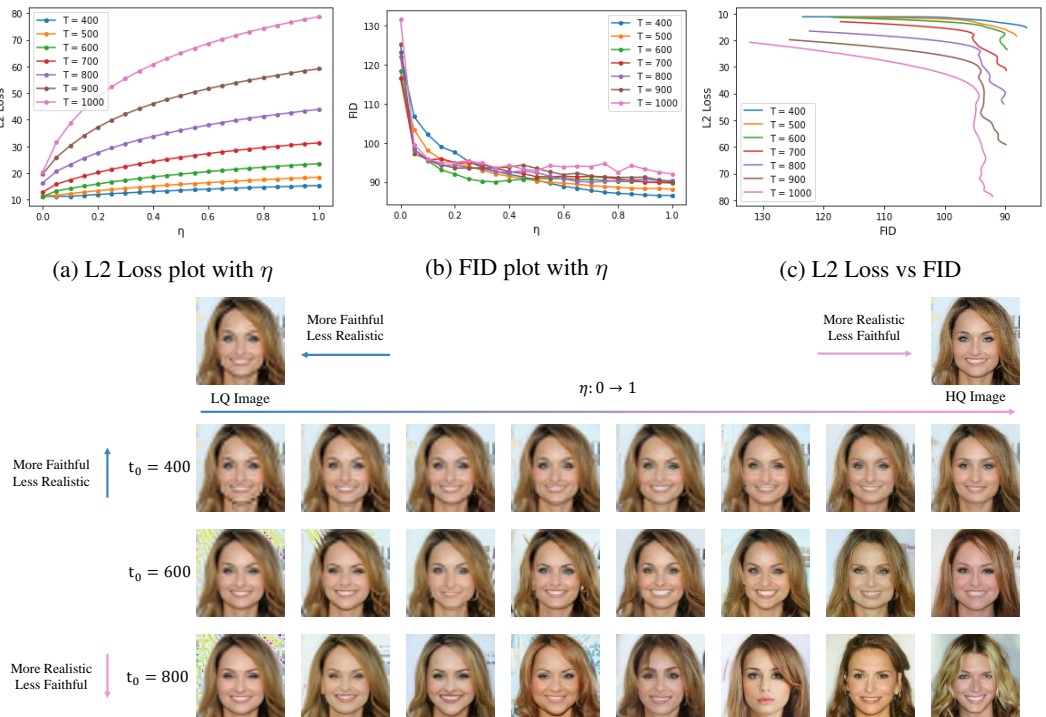

(a) L2 Loss plot with $\eta$      (b) FID plot with $\eta$      (c) L2 Loss vs FID

(d) Visualization of generated images with various hyperparameters.

Figure 11: The Faithfulness-Realism trade-off affected by $\eta$ and $t_0$ (Applying only DA Inversion).

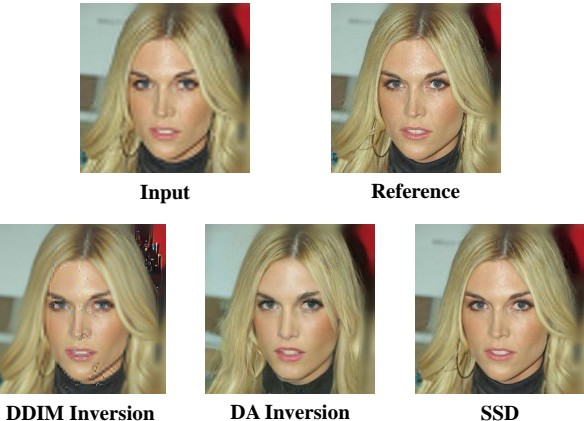

Figure 12: Visual comparison for ablation study. Performing DDIM Inversion leads to faithful but unrealistic results, while performing DA Inversion only produces faithful and realistic results. And SSD with additional consistency constraints during the generation process generates best results

| Method | PSNR (↑) | SSIM (↑) | FID (↓) | LPIPS (↓) |
|---|---|---|---|---|
| **DDIM Inversion alone** | 18.24 | 0.448 | 107.04 | 0.558 |
| **DA Inversion alone** | 23.54 | 0.663 | 41.53 | 0.321 |
| **SSD** | 28.74 | 0.816 | 32.45 | 0.202 |

Table 6: Quantitative evaluation for ablation study.

## H    ADDITIONAL RESULTS

We provide restoration results of SSD on SR tasks at various scales (ranging from $2\times$ to $32\times$) using an average-pooling downsampler, as depicted in Fig. 14. These results demonstrate the ability of SSD to handle diverse degrees of degradation. We also extended inpainting task on CelebA datasets, results can be found in Table. 7 and Fig. 13

| CelebA | Inpaint(Random) | Inpaint(Box) | NFEs↓ |
|---|---|---|---|
| Method | FID↓ / LPIPS↓ | FID↓ / LPIPS↓ | |
| DDRM | 18.09 / 0.111 | 9.75 / 0.071 | 100 |
| DPS | 30.43 / 0.259 | 30.08 / 0.253 | 100 |
| DDNM | 11.47 / 0.082 | 10.17 / 0.062 | 250 |
| SSD | 11.41 / 0.094 | 5.56 / 0.053 | 100 |

Table 7: Quantitative evaluation on the **CelebA** datasets for Inpaint task. Red indicates the best performance. For inpaint(random) task, we randomly drop the pixels in the image with a probability of 0.5. For inpaint(box) task, we randomly mask images with a 96x96 rectangle.

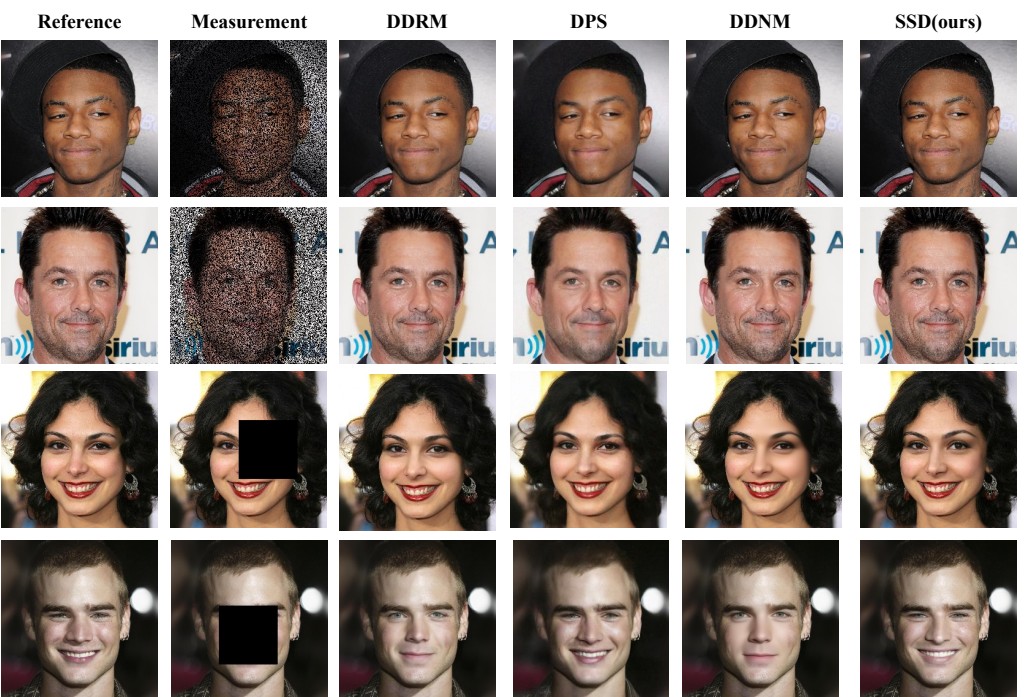

Figure 13: Inpainting results on CelebA Dataset.

Additionally, We present more qualitative results and comparisons for various IR tasks and datasets, as illustrated in Fig. 15, 16, 16, 17, 18, 19.

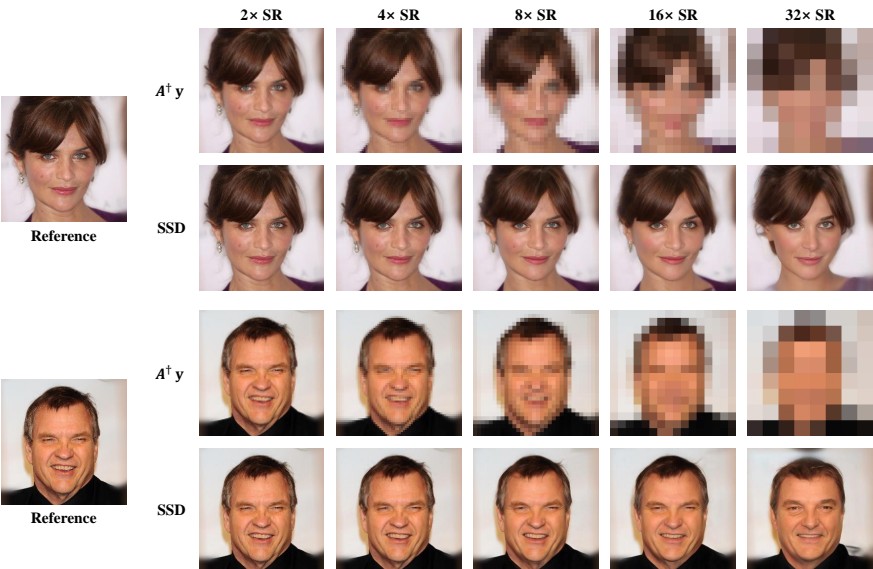

Figure 14: Restoration results at different average-pooling downsampling scales on Celeba Dataset

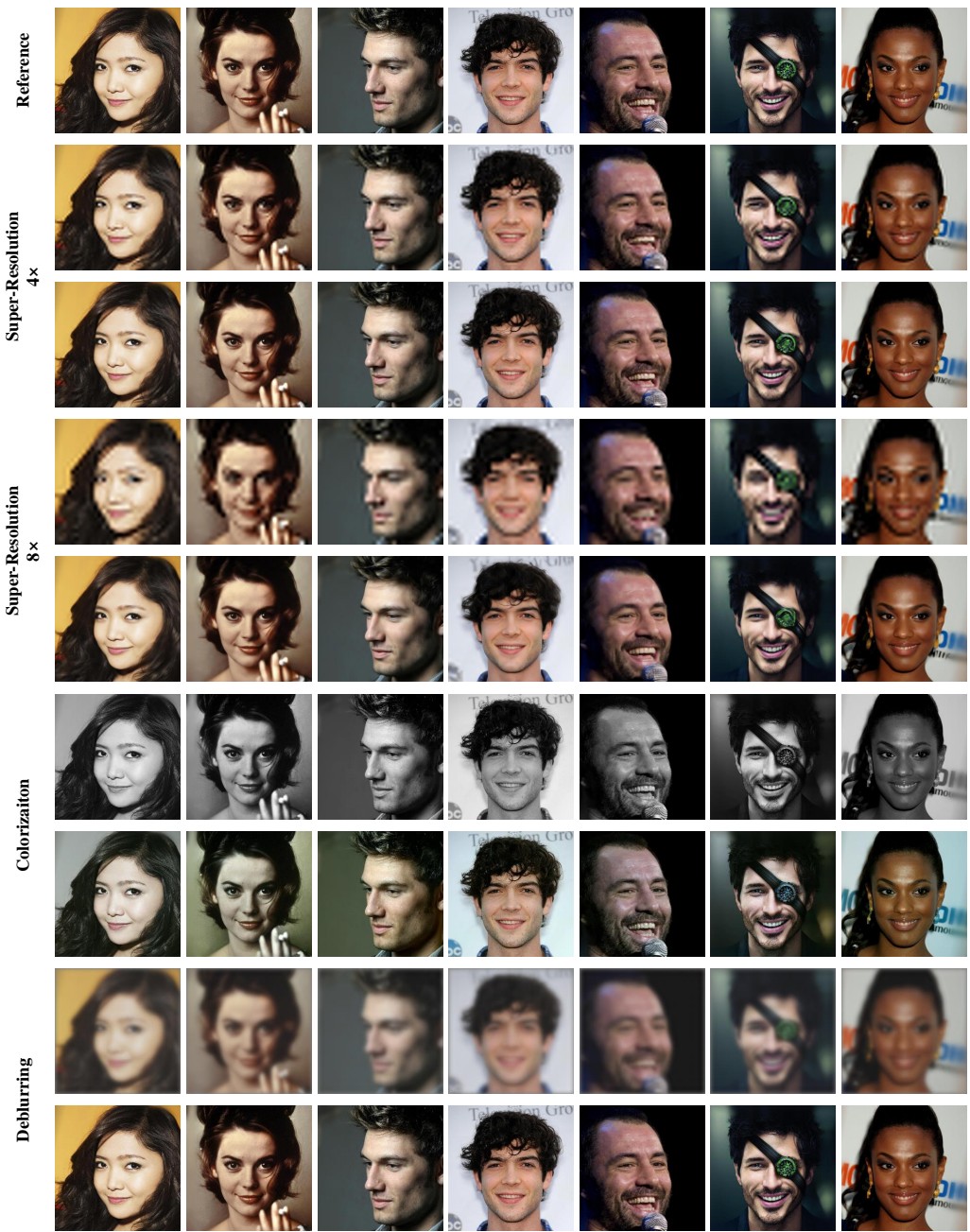

Figure 15: Image restoration results of SSD On CelebA Dataset

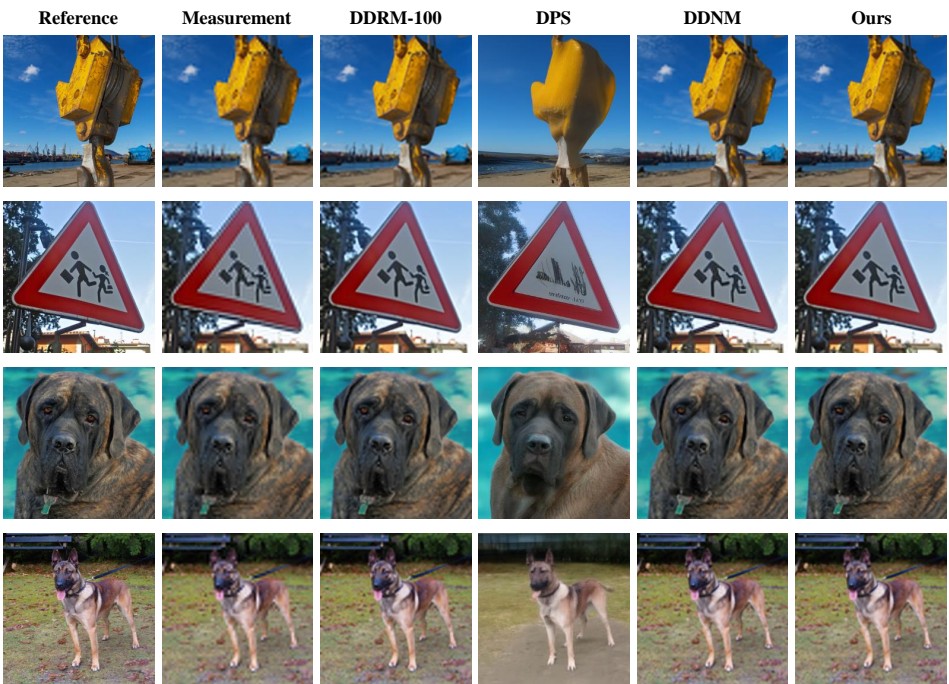

Figure 16: Comparison of results on 4× super-resoultion task using different zero-shot IR methods on the Imagenet dataset

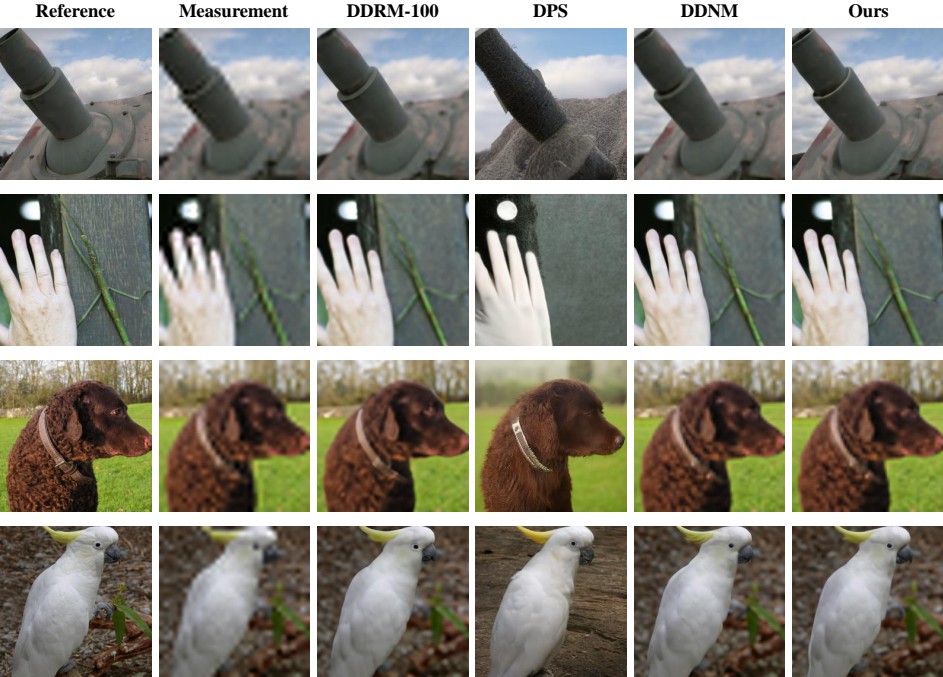

Figure 17: Comparison of results on 8× super-resoultion task using different zero-shot IR methods on the Imagenet dataset

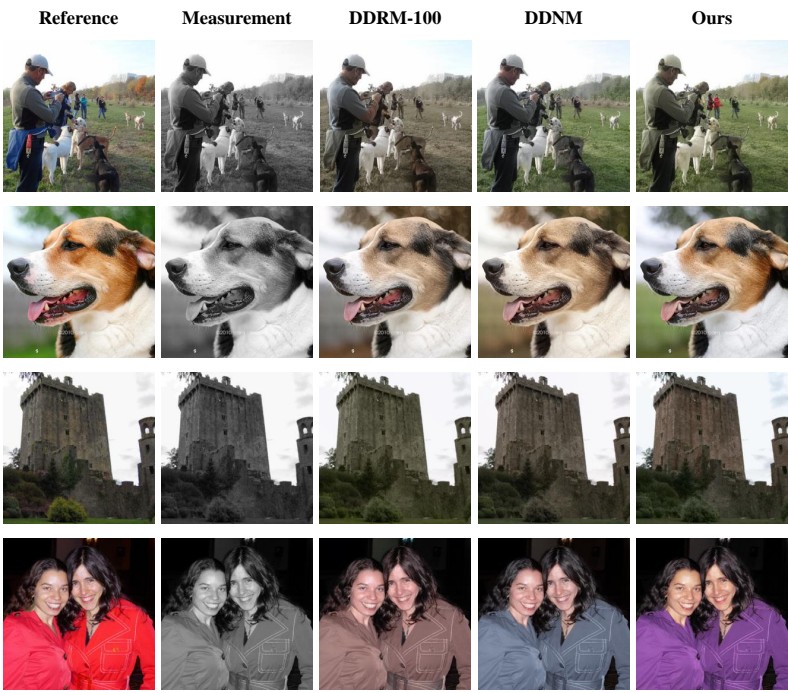

Figure 18: Comparison of results on colorization task using different zero-shot IR methods on the Imagenet dataset

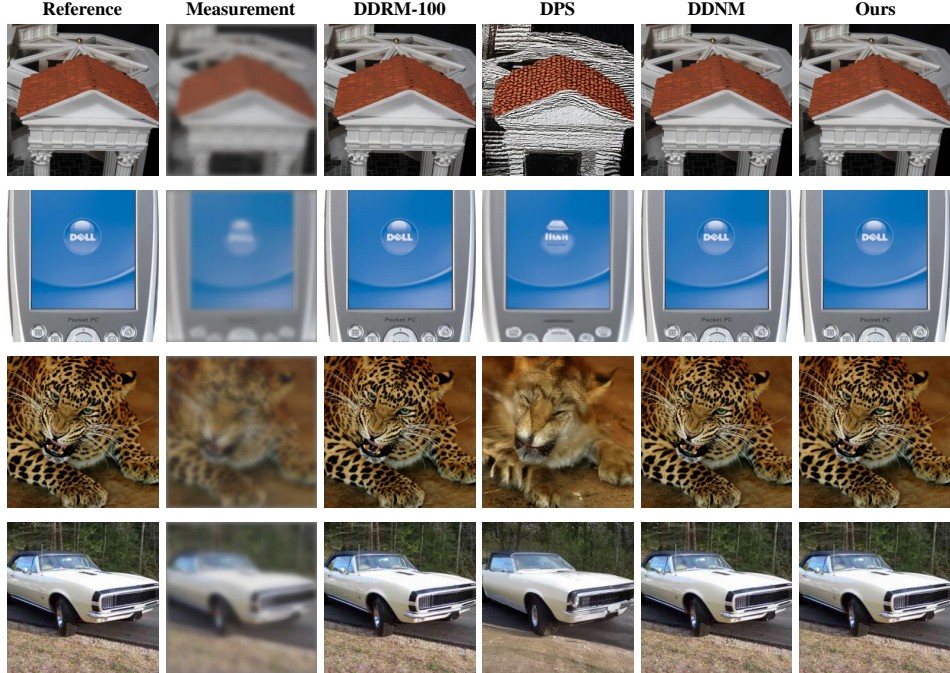

Figure 19: Comparison of results on deblurring task using different zero-shot IR methods on the Imagenet dataset

