# OpenReview forum: "Accelerating Diffusion Models for Inverse Problems through Shortcut Sampling"
_ICLR.cc/2024/Conference — ICLR 2024 Conference Withdrawn Submission_

### Official Review · Reviewer_7SJJ · 2023-10-17

**Soundness:** 2 fair
**Presentation:** 3 good
**Contribution:** 2 fair
**Rating:** 5
**Confidence:** 4

**Summary:**

The paper proposes the Shortcut Sampling for Diffusion (SSD) to solve inverse problems. The proposed Distortion Adaptive Inversion (DA Inversion) preserves the overall layout and structure of the input data. Meanwhile, the authors propose the back projection, which applies additional consistency constraints to enhance faithful. Experiments show the effectiveness of SSD regarding performance and NFEs.

**Strengths:**

1. The introduction of DA Inversion achieves a trade-off between realism and faithfulness while reducing the number of steps (NFEs).
2. The paper's structure is clear and easy to understand.

**Weaknesses:**

1. The novelty of the paper is not enough, as the back projection is mentioned in DDNM (Eqs. 13 and 14).
2. In the ablation study (Table 5), different η has little impact on performance. For PSNR and LPIPS, the gaps between η=0.1 and η=1.0 are 0.03 and 0.001respectively. This result is inconsistent with the analysis in Sec. 3.2. For example, when η=1.0, it is DDPM Inversion, but the PSNR is better than η=0.4. This cannot reflect the effectiveness of the proposed method. An explanation is needed.

**Questions:**

1. Clarify the innovation of back projection (difference from DDNM).
2. It is recommended to place the ablation study in the main paper and further analyze the results of Table 5.
3. There is a typo in Table 1 (last row): "SSD-100 (ours)" should be corrected to "SSD-30."

---

### Official Review · Reviewer_Fzfq · 2023-10-30

**Soundness:** 3 good
**Presentation:** 3 good
**Contribution:** 2 fair
**Rating:** 5
**Confidence:** 4

**Summary:**

This paper proposes a pipeline for solving inverse problems. The main idea behind this is to find an intermediate state that bridges the gap between the input measurement image and the target restored image. By utilizing this shortcut path of "Input-Embryo-Target", the proposed method achieves efficient and precise restoration results with reduced steps. The paper also introduces Distortion Adaptive Inversion for obtaining the Embryo in the inversion process, and back projection as additional consistency constraints during the generation process. The effectiveness of SSD is experimentally demonstrated on various inverse problems.

**Strengths:**

1) The task of efficient and precise restoration in inverse problems is meaningful.

2) The experimental evaluation of various inverse problems demonstrates the effectiveness of SSD.

3) The paper is overall well-structured and clearly presents the proposed methodology, including the inversion process, generation process, and the use of back projection. The figures and equations are helpful in understanding the concepts.

**Weaknesses:**

- The novelty of the paper. It seems that the core of this work is very similar to the previous work DDNM. Besides, the results are also very similar to DDNM in terms of PSNR and LIPIPS, and the improvement is marginal. This raises doubts about the effectiveness of the method.
- The author claims the proposed framework can reduce the inference step. However, in Table 1, the same or even more inference steps are used.  Besides, do the steps of inversion count into the inference steps?
- To evaluate the perceptual quality of the generated images, I recommend using some recent IQA metrics, e.g.,  CLIPIQA, and MUSIQ.
- More ablation studies should be provided, especially to demonstrate the necessity of the proposed DA Inversion. The existing results in Fig. 11 are not enough to demonstrate its necessity from my perspective.
- Missing recent work that shares a similar idea of skipping unnecessary steps to speed up the inference process, e.g., [a] [b]

[a] ExposureDiffusion: Learning to Expose for Low-light Image Enhancement, ICCV23
[b] ResShift: Efficient Diffusion Model for Image Super-resolution by Residual Shifting

**Questions:**

Please see the weakness part

---

### Official Review · Reviewer_X46h · 2023-10-31

**Soundness:** 3 good
**Presentation:** 3 good
**Contribution:** 3 good
**Rating:** 5
**Confidence:** 2

**Summary:**

This paper presents the Shortcut Samplng for Diffusion (SSD) method for solving inverse problems. SSD aims to find the "Embryo", a transitional state that bridges the measurement image y and the restored image x, which offers precise and fast restoration. The Distortion Adaptive Inversion is proposed to obtain this Embryo and the back projection and attention injection are applied for obtaining more consistent generation results. The extensive experiments demonstrate the effectiveness of SSD on several representative tasks.

**Strengths:**

1. This paper highlights a critical challenge: how to enhance images through an inversion-based approach, building a more meaningful mapping between the latent to connect degraded images and high-quality ones while preserving the consistency.
2. The idea of distortion adaptive inversion and back projection well align with conventional signal processing concepts and methods and make sense.
3. Some experimental results show excellent performance.

**Weaknesses:**

1. There are numerous aspects of the experiments that should be reviewed:
a) The proposed method can achieve superior performance in FID but PSNR results seem to be not good. For colourization tasks, I think the FID value can represent some critical factors of image quality, while for SR and deblurring, the PSNR and LPIPS (measured on each sample) might be more convincing.
b) Some GAN/Diffusion-based zero-shot restoration methods should be also compared:
[1] Xingang Pan, Xiaohang Zhan, Bo Dai, Dahua Lin, Chen Change Loy, and Ping Luo. "Exploiting Deep Generative Prior for Versatile Image Restoration and Manipulation," ECCV, 2020.
[2] Fei, Ben, Zhaoyang Lyu, Liang Pan, Junzhe Zhang, Weidong Yang, Tian-jian Luo, Bo Zhang, and Bo Dai, "Generative Diffusion Prior for Unified Image Restoration and Enhancement," CVPR, 2023.

2. For the visual result comparisons, it is hard to say which method is better. For example, in Fig. 4, for the dog case, DPS's result is obviously much superior to the proposed one.

3. SSD relies on an accurate estimation of degraded operators. For example, "due to SSD relies on an accurate estimation of degraded operators" should be "due to SSD's reliance relies on an accurate estimation of degraded operators.

**Questions:**

Please see weakness.

---

### Official Review · Reviewer_4hTS · 2023-11-06

**Soundness:** 2 fair
**Presentation:** 3 good
**Contribution:** 2 fair
**Rating:** 3
**Confidence:** 3

**Summary:**

The paper targets the so called “inverse problem”, which is quite prominent in Diffusion model literature and have been in limelight for last few years. The authors of this paper proposes “Shortcut Sampling”, a specific way of solving inverse problem. The core idea is to not start from noise when simulating posterior sampling. Instead, the author proposes to start from a middle state — termed “Embryo”. From the middle-state, the authors followed the usual generation and back-projection technique.

The authors showed their method to be on par or sometimes better than others while solving popular inverse problems like super-res, colorization, deblurring.

**Strengths:**

The problem targeted by the paper is of high importance in commercial applications. The proposed method has a good motivation and has merit in terms of its conceptual offering. At a high level, the proposal does make sense, i.e. it is indeed reasonable to not start the posterior sampling from pure noise. Some of the results are encouraging.

**Weaknesses:**

Even though I agree that the proposed idea has a good motivation, its technical details are unclear or questionable to me. The paper is overall well written but some of its confusion math notations made it even harder for me to assess the technical correctness.

- I did not really understand the reason behind Eq. 10. Did this come from a prior work ? Why a $\beta_{t+1}$ appeared suddenly ? There is virtually no explanation (conceptual or mathematical) around Eq.10. Authors said “we can define a similar form ..” which is not a very solid reasoning.
- Even though there is no reference to supplementary, but it seems the relevant part of the explanation for Eq.10 is in appendix C. The derivation and reasoning in app.C is also questionable. What I could understand from app.C is that the authors derived Eq.33 from the **forward process** and tried to conclude the DA inversion equation (Eq. 10) should look similar. Is this even theoretically reasonable ?
- It seems to me that the authors are trying to compare with noise $\epsilon$ with noise-estimate $\epsilon_{\theta^*}(x_t, t)$, which is not sometimes one can do. A trained model $\epsilon_{\theta^*}$ can have very different statistics than $\mathcal{N}(0, I)$. Do the authors agree ?

Notations issues:

- 3rd paragraph of intro: Can you please properly define $z$ before using it ? Generally in inverse problems, posterior is shown as $p(x|y)$ and it is confusing what $z$ is and how to interpret it. If $z$ is just prior (i.e. $\mathcal{N}(0, I)$), then the notation $p(z|y)$ looks very strange !
- Just after Eq. 1, write discrete sequences as $[x_t]\_{t=0}^T$ and not ${x_t}_{t=0}^T$. Same for $\beta_t$.
- The notation $\epsilon$ and $\bar\epsilon$ are very confusing. I am not sure what means whats. Sometimes they are defined as just gaussian noise (also same as $\epsilon$), sometimes as “noise added until ..”. (this definition only appears in supplementary). What exactly does that mean ? It is never mathematically defined or explained.
- Eq. 10 and Eq. 12: $z$ has two different definition. Where did $(\mu, \sigma)$ come from in Eq. 12 ?
- No definition for $H^{\dagger}$ where it’s first defined.

**Questions:**

See weakness for consolidated questions and comments.